# Crosslinking by ZapD drives the assembly of short FtsZ filaments into toroidal structures in solution

Adrián Merino-Salomón[1], Jonathan Schneider[2], Leon Babl[1], Jan-Hagen Krohn[1,3], Marta Sobrinos-Sanguino[4], Tillman Schäfer[5], Juan Ramon Luque-Ortega[4], Carlos Alfonso[6], Mercedes Jiménez[6], Marion Jasnin[7,8]*, Petra Schwille[1]*, Germán Rivas[6]*

[1]Department of Cellular and Molecular Biophysics, Max Planck Institute of Biochemistry, Martinsried, Germany; [2]Department of Molecular Structural Biology, Max Planck Institute of Biochemistry, Martinsried, Germany; [3]Exzellenzcluster ORIGINS, Garching, Germany; [4]Molecular Interactions Facility, Centro de Investigaciones Biológicas Margarita Salas, Consejo Superior de Investigaciones Científicas (CSIC), Madrid, Spain; [5]Cryo-EM Facility, Max Planck Institute of Biochemistry, Martinsried, Germany; [6]Centro de Investigaciones Biológicas Margarita Salas, Consejo Superior de Investigaciones Científicas (CSIC), Madrid, Spain; [7]Helmholtz Pioneer Campus, Helmholtz Munich, Neuherberg, Germany; [8]Department of Chemistry, Technical University of Munich, Garching, Germany

*For correspondence:
marion.jasnin@helmholtz-munich.de (MJ);
schwille@biochem.mpg.de (PS);
grivas@cib.csic.es (GR)

Competing interest: The authors declare that no competing interests exist.

## eLife Assessment

The formation of the Z-ring at the time of bacterial cell division interests researchers working towards understanding cell division across all domains of life. The manuscript by Jasnin et al reports the cryoET structure of toroid assembly formation of FtsZ filaments driven by ZapD as the cross linker. The findings are **important** and have the potential to open a new dimension in the field, and the evidence to support these exciting claims is **solid**.

**Abstract** Cell division in *Escherichia coli* relies on the Z ring, a cytoskeletal structure that acts as a scaffold for the assembly of the divisome. To date, the detailed mechanisms underlying the assembly and stabilization of the Z ring remain elusive. This study highlights the role of the FtsZ-associated protein (Zap) ZapD in the assembly and stabilization of Z-ring-like structures via filament crosslinking. Using cryo-electron tomography and biochemical analysis, we show that, at equimolar concentrations of ZapD and FtsZ, ZapD induces the formation of toroidal structures composed of short, curved FtsZ filaments that are crosslinked vertically, but also laterally and diagonally. At higher concentrations of ZapD, regularly spaced ZapD dimers crosslink FtsZ filaments from above, resulting in the formation of straight bundles. Despite the simplicity of this reconstituted system, these findings provide valuable insights into the structural organization and stabilization of the Z ring by Zap proteins in bacterial cells, revealing the key role of optimal crosslinking density and geometry in enabling filament curvature and ring formation.

## Introduction

Cell division in bacteria is a well-orchestrated process mediated by a multiprotein complex called the divisome. Their components interact reversibly to form a division ring, which is essential for cytokinesis (*Levin and Janakiraman, 2021*; *Du and Lutkenhaus, 2019*; *Attaibi and den Blaauwen, 2022*). At the core of this process in most bacteria is FtsZ, a tubulin homolog that polymerizes in the presence of GTP beneath the inner membrane at the future division site, providing a scaffold for recruiting other division proteins (*Nogales et al., 1998*; *de Boer et al., 1992*; *Addinall and Lutkenhaus, 1996*; *McQuillen and Xiao, 2020*; *Wang et al., 2020*). FtsZ undergoes treadmilling due to polymerization-dependent GTP hydrolysis, allowing the ring to exhibit its dynamic behavior (*Wagstaff et al., 2017*). Depending on environmental conditions, FtsZ filaments can assemble into various structures, including bundles, sheets, and toroids (*Lu et al., 2000*; *Mukherjee and Lutkenhaus, 1999*; *Löwe and Amos, 1999*; *González et al., 2003*; *Popp et al., 2009*). Membrane-tethered FtsZ assemblies are arranged as loosely structured spirals in the midcell region, guided by spatial regulators. These structures eventually coalesce to form a cohesive Z ring (*Squyres et al., 2021*).

The molecular components involved in cell division must be in the right place at the right time during the cell cycle. Several factors in this spatiotemporal regulation modulate FtsZ assembly. In *Escherichia coli*, FtsA and ZipA tether FtsZ to the membrane (*Ma and Margolin, 1999*; *Hale and Boer, 2002*; *Wang et al., 1997*) and, together with a set of FtsZ-associated proteins (Zaps), stabilize the ring (*Durand-Heredia et al., 2012*; *Huang et al., 2013*; *Durand-Heredia et al., 2011*; *Ebersbach et al., 2008*). Conversely, negative modulators inhibit FtsZ assembly at other sites, namely Min proteins at the polar regions (*de Boer et al., 1989*) and SlmA on the nucleoid (*Tonthat et al., 2011*). Most of these modulators interact with FtsZ through its carboxy-terminal end (CCTP), which modulates division assembly as a central hub (*Ortiz et al., 2016*; *Buske and Levin, 2012*). ZapD is the only Zap protein known to crosslink FtsZ by binding CCTP, suggesting a critical Z ring structure stabilizing function (*Durand-Heredia et al., 2012*; *Schumacher et al., 2016*; *Huang et al., 2016*; *Choi et al., 2016*; *Roach et al., 2016*; *Figure 1a*).

To date, a high-resolution structure of the Z ring remains unresolved. Most evidence suggests that FtsZ filaments are arranged in short patches along the equatorial circle, held together by lateral interactions and interactions with Zap proteins (*Du and Lutkenhaus, 2019*; *McQuillen and Xiao, 2020*; *Huecas et al., 2017*; *Sundararajan and Goley, 2017*; *Szwedziak et al., 2014*). The dynamics of treadmilling and lateral interactions between FtsZ filaments, as well as crosslinking by FtsZ-associated proteins (Zaps), are thought to play a crucial role in the condensation of the Z ring (*Whitley et al., 2021*). Specifically, the crosslinking of filaments by Zaps, particularly ZapD, is vital for maintaining the neighboring organization of the filaments into a ring structure. This organization may also be critical for the functionality of the divisome and the mechanism of force generation during cytokinesis (*Levin and Janakiraman, 2021*; *Du and Lutkenhaus, 2019*; *McQuillen and Xiao, 2020*; *Söderström and Daley, 2017*; *Erickson et al., 2010*; *Nguyen et al., 2021*; *Mateos-Gil et al., 2019*).

Zap proteins play overlapping roles in stabilizing the Z ring during cell division. Although they are nonessential for division, their absence leads to less compact Z rings, and the knowout of two or more Zap encoding genes simultaneously results in cell elongation (*Durand-Heredia et al., 2012*; *Buss et al., 2013*; *Hale et al., 2011*). Notably, these different Zap proteins do not share any sequence homology and differ in their structure and filament crosslinking mechanism (*Huang et al., 2013*; *Schumacher et al., 2016*; *Schumacher et al., 2017*). Additionally, Zap proteins may actively remove FtsZ from the septum during the final stages of constriction (*Pazos et al., 2013*). They also facilitate the positioning of the Z ring at the replication terminus of the chromosome by interacting with MatP, which forms a scaffolding anchor known as Ter-linkage (*Bailey et al., 2014*).

ZapD is a symmetrical dimer consisting of an α-helical domain and a β-sheet domain containing a positively charged binding pocket required for crosslinking and bundling FtsZ filaments (*Roach et al., 2016*; *Schumacher et al., 2017*). ZapD binds to the CCTP of FtsZ through electrostatic interactions (*Roach et al., 2016*). ZapD crosslinks FtsZ filaments, promoting bundling and significantly reducing the GTPase activity of FtsZ (*Durand-Heredia et al., 2012*). A model of how ZapD crosslinks neighboring FtsZ filaments has been proposed based on the structural analysis of ZapD-FtsZ interactions (*Schumacher et al., 2017*). In this model, the ZapD dimer connects two FtsZ filaments through the CCTP, which could organize them in a parallel or antiparallel orientation due to the flexibility provided by the flexible linker (*Figure 1a*). However, the experimental validation of this model is still missing.

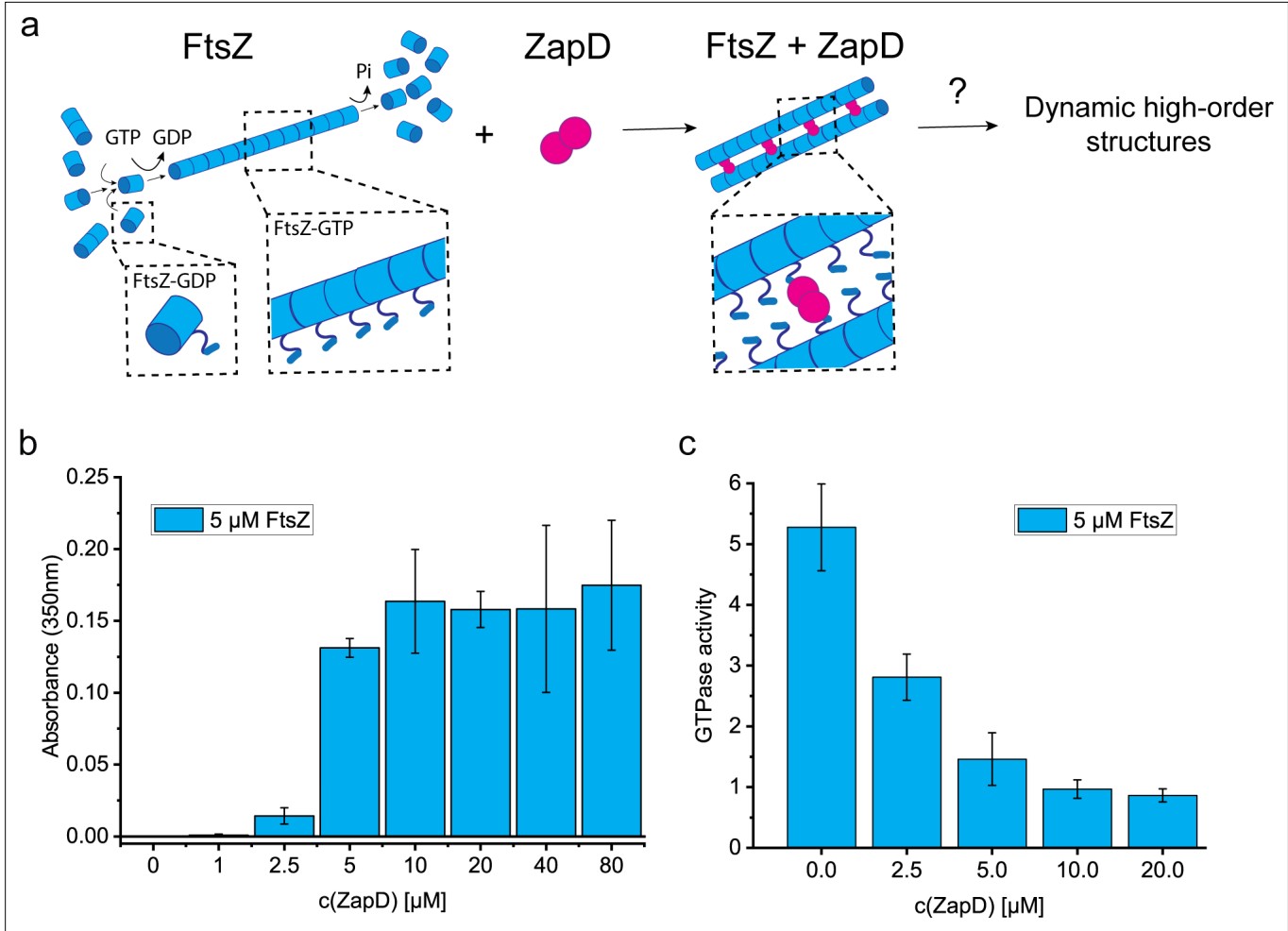

**Figure 1.** ZapD binds FtsZ and promotes filament bundling. (**a**) Scheme of the FtsZ protein and its interaction with ZapD. *E. coli* FtsZ (blue) monomers in solution oligomerize depending on the buffer conditions. Upon GTP binding, FtsZ homopolymerizes directionally, assembling single-stranded filaments. ZapD is a robust dimer (magenta) that interacts directly with FtsZ, crosslinking filaments by promoting lateral interactions between them. Although the molecular mechanism is still unclear, the current hypothesis of interaction assumes dimers of ZapD crosslinking two FtsZ filaments through the CCTP of FtsZ, expecting at around 1:1 (FtsZ:ZapD) molar ratio in a homogeneous bundle (1 dimer of ZapD connected to 2 monomers of FtsZ). According to this model, the orientation of the FtsZ filaments could be parallel or antiparallel, allowing the growth and treadmilling of the filaments. However, the mechanism of assembly of dynamic high-order structures is still unknown. (**b**) Turbidity assays measuring the absorbance at 350 nm of samples containing 5 μM FtsZ and increasing concentrations of ZapD. The turbidity of the sample was measured 5 min after the addition of 1 mM GTP at working buffer (50 mM KCl, 50 mM Tris-Cl, 5 mM MgCl$_2$, pH 7). FtsZ polymers do not show a significant turbidity at this wavelength; therefore, the signal at 350 nm corresponds to the presence of large FtsZ macrostructures and bundles. The mean value and SD of >3 independent replicates are plotted in the graph. (**c**) GTPase activity of FtsZ after the addition of 1 mM GTP in the presence of increasing concentrations of ZapD at working conditions (50 mM KCl, 50 mM Tris-Cl, 5 mM MgCl$_2$, pH 7). The mean value and SD plotted in the graph are the result of 3 independent replicates. The GTPase activity was measured as a result of the Pi released from GTP consumption. The units are mol GTP consumed per mol FtsZ per min.

The online version of this article includes the following source data and figure supplement(s) for figure 1:

**Figure supplement 1.** Characterization of the ZapD dimer and ZapD-FtsZ-GDP interaction by analytical ultracentrifugation (AUC*)*.

**Figure supplement 2.** Biochemical characterization of the ZapD-FtsZ-GDP interaction by fluorescence correlation microscopy (FCS) and fluorescence anisotropy.

**Figure supplement 3.** Biochemical characterization of the ZapD and FtsZ bundles by turbidity at 350 nm.

**Figure supplement 4.** Ionic strength in the buffer lowers the effect of ZapD over FtsZ GTPase activity.

**Figure supplement 5.** ZapD binding to FtsZ polymers via sedimentation assays.

**Figure supplement 5—source data 1.** Raw images of the SDS-PAGE gels used in *Figure 1—figure supplement 5* and for quantification, corresponds to the raw SDS-PAGE gels.

**Figure supplement 5—source data 2.** Raw images of the SDS-PAGE gels used in *Figure 1—figure supplement 5* and for quantification, includes the labelled images.

Furthermore, the link between FtsZ bundling, promoted by ZapD, and the large-scale organization of FtsZ filaments remains unresolved. Therefore, investigating how ZapD crosslinks FtsZ filaments may shed light on the molecular mechanisms underlying Z ring assembly. Recently, *Gong et al., 2024*, demonstrated that although the FtsZ crosslinking proteins are not individually essential, they collectively play a critical role in the condensation and stability of the Z ring in different organisms. However, the specific mechanisms driving this condensation remain unclear, highlighting the complexity of the bacterial cell division process.

In this study, we used cryo-electron microscopy (cryo-EM) and cryo-electron tomography (cryo-ET), together with biochemical and biophysical assays, to investigate the structural organization of FtsZ polymers in the presence of ZapD. This integrated approach revealed that, at equimolar concentrations of ZapD and FtsZ, ZapD facilitates the assembly of FtsZ filaments into toroidal polymers in solution. This observation is consistent with the low-resolution structure of the Z ring obtained in vivo (*McQuillen and Xiao, 2020*; *Fu et al., 2010*; *Li et al., 2007*; *Lyu et al., 2016*). Our results allowed us to propose a molecular mechanism for toroid formation, providing valuable insights into how crosslinking of division proteins drives the stabilization of the Z ring.

## Results

### ZapD dimer interacts with FtsZ-GDP oligomers

Previous structural studies have identified a direct interaction between ZapD and FtsZ through the CCTP or central hub of FtsZ, involving charged residues in the interaction (*Huang et al., 2016*; *Roach et al., 2016*). To independently verify these findings, we investigated the interaction between FtsZ-GDP and ZapD using analytical ultracentrifugation (AUC) (see *Figure 1—figure supplement 1*). First, we confirmed that ZapD forms stable dimers, which is consistent with previous studies (*Durand-Heredia et al., 2012*; *Huang et al., 2016*; *Schumacher et al., 2017*; *Figure 1—figure supplement 1aI and b*). In contrast, unpolymerized FtsZ-GDP primarily self-assembles from monomers into dimers and other oligomeric species, albeit in small proportions, as expected (*Rivas et al., 2000*; *Figure 1—figure supplement 1aII*).

Sedimentation velocity analysis of mixtures of the two proteins revealed the presence of two predominant molecular species of ZapD:FtsZ complexes in solution. These complexes are compatible with ZapD dimers bound to one or two FtsZ monomers, corresponding to ZapD:FtsZ stoichiometries of 2:1 and 1:1, respectively (*Figure 1—figure supplement 1aIII–IV*). This observation is consistent with the proposed interaction model (*Schumacher et al., 2017*). Furthermore, we confirmed the interaction of both proteins using fluorescence anisotropy and fluorescence correlation spectroscopy (*Figure 1—figure supplement 2a and b*). In these experiments, an increase in fluorescence anisotropy or diffusion time suggested the formation of larger particles due to protein-protein interactions.

### ZapD binding promotes the bundling of FtsZ-GTP polymers

ZapD is a crosslinker for FtsZ filaments, promoting their bundling in solution (*Durand-Heredia et al., 2012*). However, the precise molecular mechanism by which ZapD crosslinks the filaments remains unclear. To investigate the biochemical features underlying this bundling process, we examined how the formation of large FtsZ bundles or higher-order FtsZ polymer structures depends on the concentration of ZapD. For this analysis, we used turbidity measurements at 350 nm. When testing GTP-FtsZ filaments at 5 or 10 µM concentrations, we found negligible turbidity, similar to that of ZapD alone at concentrations ranging from 1 to 40 µM. However, when ZapD was added, the turbidity of the FtsZ polymers increased, reaching a maximum at approximately 5 µM of ZapD (with a ZapD-FtsZ molar ratio of around 1). This turbidity level did not change significantly even at higher ZapD concentrations (*Figure 1b*, *Figure 1—figure supplement 3*).

In the presence of ZapD, FtsZ rapidly forms higher-order polymers upon addition of GTP, as indicated by turbidity assays. In contrast, the formation of single- or double-stranded FtsZ filaments without ZapD does not result in a significant increase in turbidity. Compared to FtsZ filaments observed in vitro, FtsZ bundles with ZapD exhibited reduced GTPase activity, reaching 20% at equimolar or higher concentrations of ZapD (*Figure 1c*, *Figure 1—figure supplement 4*), in agreement with previous studies (*Durand-Heredia et al., 2012*). The macrostructures formed by FtsZ in the presence of ZapD were disassembled more slowly than the FtsZ filaments upon GTP consumption (*Figure 1—figure*

*supplement 3*). Replenishment of GTP allowed for a partial recovery of the turbidity signal, suggesting that the FtsZ bundles can be rapidly reassembled (*Figure 1—figure supplement 3e*). Previous studies have shown that high concentrations of macromolecular crowders (such as Ficoll or dextran) promote the formation of dynamic FtsZ polymer networks (*González et al., 2003*), observations that are consistent with our results. In these cases, the GTPase activity of FtsZ was significantly reduced compared to that of FtsZ filaments, leading to a decrease in GTPase turnover. Similar mechanisms may apply to assembly reactions involving ZapD (*Durand-Heredia et al., 2012*; *Huang et al., 2016*).

To understand the relationship between the concentration dependence of the turbidity signal – associated with the formation of FtsZ bundles or higher-order structures driven by ZapD (see *Figure 1b*) – and the amount of ZapD bound to the FtsZ binding sites, we carried out analytical centrifugation sedimentation velocity assays, previously used to characterize the binding of ZapD to GDP-FtsZ. This method allowed us to separate ZapD-FtsZ bundles from free ZapD in solution and calculate the concentration of unbound ZapD from the interference signal of the slowly sedimenting protein behind the rapidly sedimenting heavy ZapD-FtsZ complexes. Using the measured concentration of free ZapD together with known input concentrations of ZapD and FtsZ, we were able to calculate the concentration of bound ZapD and the binding stoichiometry of ZapD in the bundles (moles of bound ZapD in the bundles per mole of FtsZ in monomer molar units of both proteins) at different total concentrations of ZapD (see *Figure 1—figure supplement 5*). Our results showed that in the 5–10 µM ZapD concentration range, the bundles exhibited an estimated ZapD binding stoichiometry of approximately 0.3–0.4, meaning that one ZapD dimer is associated with every four to six FtsZ monomers. In contrast, at higher ZapD concentrations (30–40 µM), the binding stoichiometry increased to 1.1±0.2, indicating one ZapD dimer for every two FtsZ monomers. These results are qualitatively consistent with parallel co-sedimentation assays and additional analysis by SDS-PAGE, showing an increase in the relative intensity of ZapD in the pellet containing the FtsZ bundles at elevated ZapD concentrations in the mixture (see *Figure 1—figure supplement 5*).

These results suggest that the ZapD binding stoichiometry in the polymers increases with higher ZapD concentrations. We then carried out cryo-EM and cryo-ET studies to investigate whether this increase in binding density affects the structural organization of the bundles formed.

## ZapD facilitates the formation of higher-order FtsZ-GTP toroidal structures

Previous studies have visualized bundles with similar features using negative-stain transmission electron microscopy (*Durand-Heredia et al., 2012*; *Roach et al., 2016*; *Schumacher et al., 2017*). Here, cryo-EM allowed us to avoid any staining artifact or structural distortion (due to adsorption or flattening upon grid drying; *Kourkoutis et al., 2012*) and to resolve single filaments. In the presence of GTP, FtsZ polymerized into thin filaments of variable length and curvature (*Figure 2a*). Upon addition of equimolar amounts of ZapD, corresponding to the slightly substoichiometric ZapD binding densities described in the previous section, FtsZ filaments assembled into bundles that predominantly formed circular toroidal structures (*Figure 2b and c*, *Figure 2—figure supplement 1a*), making them a compelling study target. A close-up view of the FtsZ toroid suggests a partially ordered arrangement of the filaments in the toroidal structure (*Figure 2c*).

Quantitative analysis of the toroidal structure revealed a conserved organization, with an outer diameter on the order of the bacterial cell size (502±55 nm) (*Figure 2d*), a typical thickness of 127±25 nm (*Figure 2e*), and an average height of 93±15 nm (*Figure 2—figure supplement 1b*). By measuring the shortest and longest axes, we determined that the circularity of the structure was 0.92±0.1 and 0.85±0.1 for the outer and inner diameters, respectively (*Figure 2—figure supplement 1c*). This conserved toroidal structure could result from the intrinsic curvature of the FtsZ filaments (*Lu et al., 2000*; *Huecas et al., 2017*; *Erickson and Osawa, 2017*) stabilized by ZapD binding. The dimension of a ZapD dimer is ~7 nm along its longest axis. *Huecas et al., 2017*, estimated an interfilament distance of ~6.5–6.7 nm for toroids of *Bacillus subtilis* FtsZ. These authors also observed a difference in this distance as a function of the linker, suggesting that linker length modulates FtsZ-FtsZ interactions. We observed a similar spacing for double filaments (5.9±0.8 nm) and a longer spacing in the presence of ZapD (7.88±2.1 nm). Previous studies with ZapD have suggested that distances of 6–12 nm are possible based on the protein structure (*Schumacher et al., 2017*). Longer linkers may

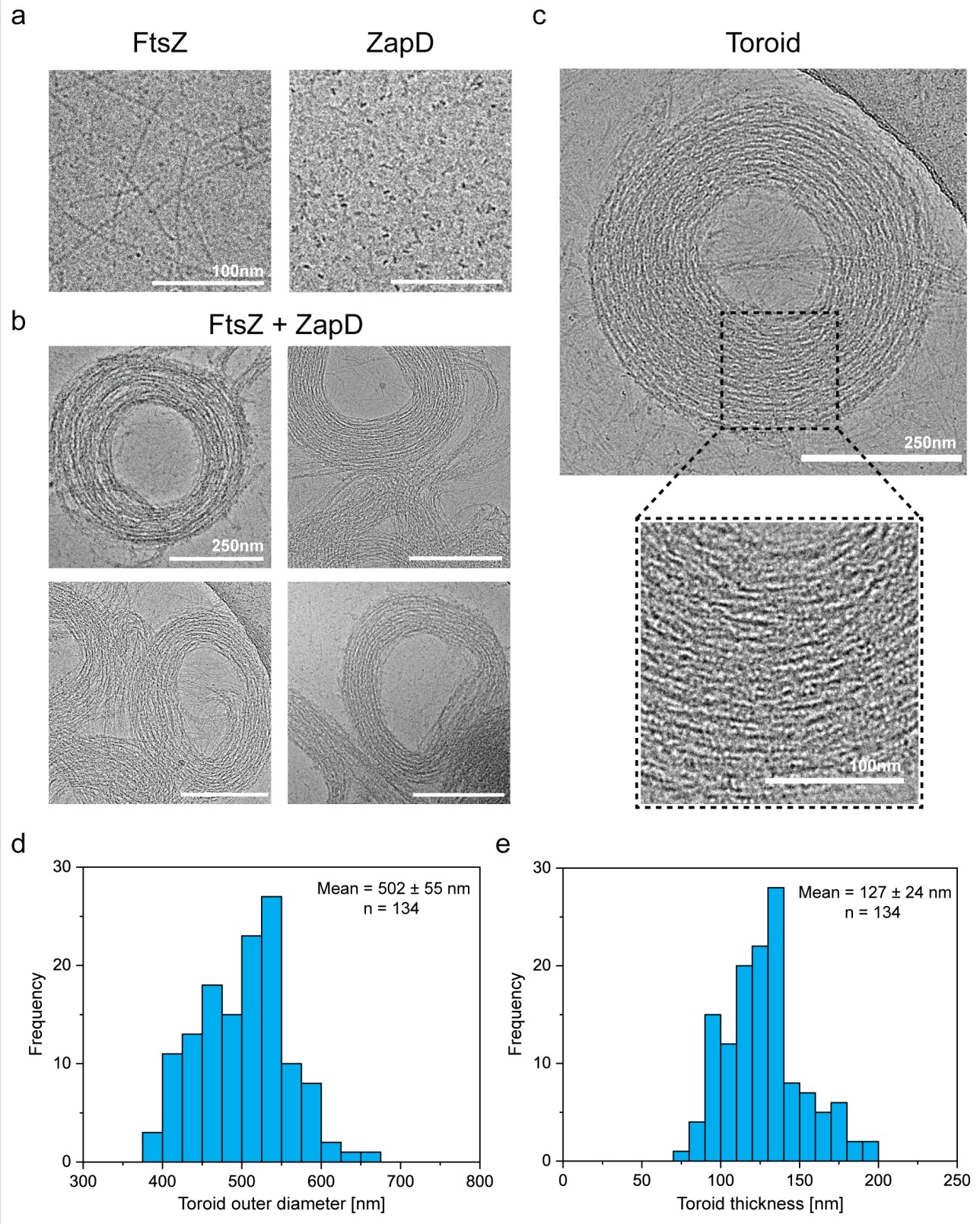

**Figure 2.** ZapD promotes the formation of FtsZ toroids. (**a**) Cryo-electron microscopy (cryo-EM) micrographs of FtsZ filaments (FtsZ-GTP form) (left) and ZapD protein (right) at 10 μM under working conditions (50 mM KCl, 50 mM Tris-Cl, 5 mM MgCl₂, pH 7). Scale bars are 100 nm. (**b**) Cryo-EM images of FtsZ (10 μM) in the presence of equimolar concentrations of ZapD (ratio 1:1) and 1 mM GTP in working conditions. Cryo-EM grids were plunge-frozen 2 min after GTP addition to favor the assembly of FtsZ and ZapD structures. The proteins were mixed before the polymerization was triggered by GTP

*Figure 2 continued on next page*

*Figure 2 continued*

addition. Scale bars are 250 nm. (**c**) Micrograph of an individual FtsZ toroid found under the same conditions as in (**b**). Close-up view of an area within the toroid is framed by a dotted black line, revealing the large amount of FtsZ filaments that form its structure. (**d**) Distribution of the outer diameter of the FtsZ toroid. Each toroid was measured perpendicularly in the shortest and longest axis to ensure the reliability of the measurement. The mean value and standard deviation are shown in the graph. (**e**) Distribution of toroidal thickness. It was measured as the result of the difference between the outer and inner diameter of each toroid. The mean value and standard deviation are shown in the graph.

The online version of this article includes the following figure supplement(s) for figure 2:

**Figure supplement 1.** FtsZ toroids and bundles formed at different protein ratios.

also provide additional freedom to spread the filaments further apart and allow a greater degree of variability in the connections made by ZapD.

Previous studies have shown different FtsZ structures at different concentrations and buffer conditions. FtsZ filaments are flexible and can generate different curvatures ranging from mini rings of ~24 nm to intermediate circular filaments of ~300 nm or toroids of ~500 nm in diameter (reviewed in *Erickson and Osawa, 2017*; *Wang et al., 2019*). It is reasonable to assume that FtsZ filaments can accommodate the toroidal shape promoted by ZapD crosslinking.

Our data showed that ZapD dimers crosslink adjacent FtsZ filaments into bundles and toroids. These toroids are remarkably regular in size and similar to the bacterial diameter (*McQuillen and Xiao, 2020*; *Fu et al., 2010*; *Li et al., 2007*), suggesting a conserved intrinsic curvature of the FtsZ filaments across a range of ZapD bonds (*Erickson and Osawa, 2017*).

## 3D structure of ZapD-mediated FtsZ toroids revealed by cryo-ET

Visualization of toroidal FtsZ structures using cryo-EM revealed a complex arrangement of FtsZ filaments within the toroidal shape (*Figure 2c*). To gain a more detailed understanding of their three-dimensional (3D) organization, we used cryo-ET. Our initial analysis focused on ZapD-mediated toroidal FtsZ structures at equimolar concentrations of ZapD and FtsZ (*Figure 3*, *Figure 3—figure supplement 1a*).

Cryo-ET revealed the dense packing of the toroidal structures, with numerous densities connecting the FtsZ filaments laterally or diagonally (*Figure 3a–c*, *Figure 3—figure supplement 1a*, and *Figure 3—figure supplement 2a*). Zoomed-in views of the toroids in the XY plane showed that the toroid consists of relatively short filament segments arranged nearly parallel or antiparallel to each other (*Figure 3b*, *Figure 3—figure supplement 2a*). In contrast, cross-sectional views of the toroids revealed elongated structures rather than simple filament cross-sections along the Z-axis (*Figure 3c*, left, *Figure 3—figure supplement 3a*).

We then extracted the toroid isosurface from the tomograms to visualize its 3D structure (*Figure 3c*, right–e, *Figure 3—figure supplement 2b–e*, and *Video 1*). The isosurface confirmed the presence of extended structures along the Z-axis, well beyond the elongation expected from the missing wedge effect for single FtsZ filaments (for comparison, see *Figure 3—figure supplement 4*). The vertically extended structures appeared to correspond to filaments that were connected or decorated by additional densities along the Z-axis (*Figure 3—figure supplement 3b*). Importantly, these densities were only observed in the presence of ZapD (*Figure 3—figure supplement 4b*), suggesting that they represent ZapD connections (*Figure 3e*, *Figure 3—figure supplements 2e and 3b*). We note that the resolution of the data is not sufficient to precisely resolve ZapD proteins from the FtsZ filaments in the Z-axis.

These results suggest that the toroids are constructed and stabilized by interactions between ZapD and FtsZ, which are mainly formed along the Z-axis but also vertically and diagonally.

## ZapD plasticity is essential for toroid shape and stabilization

Next, we manually labeled the connecting densities in the toroid isosurfaces to analyze their arrangement and connectivity with the FtsZ filaments (*Figure 4a*; *Video 2*). The high density of the toroids and the wide variety of conformations of these densities prevented the use of subtomogram averaging to resolve their structure and spatial arrangement within the toroids. Most connections exhibited a characteristic bi-spherical shape between the filaments, reminiscent of ZapD dimers (*Figure 4b–d*). We observed lateral connections between two parallel FtsZ filaments at the same height within the toroid (*Figure 4b*). In addition, FtsZ filaments at different heights can connect vertically and diagonally

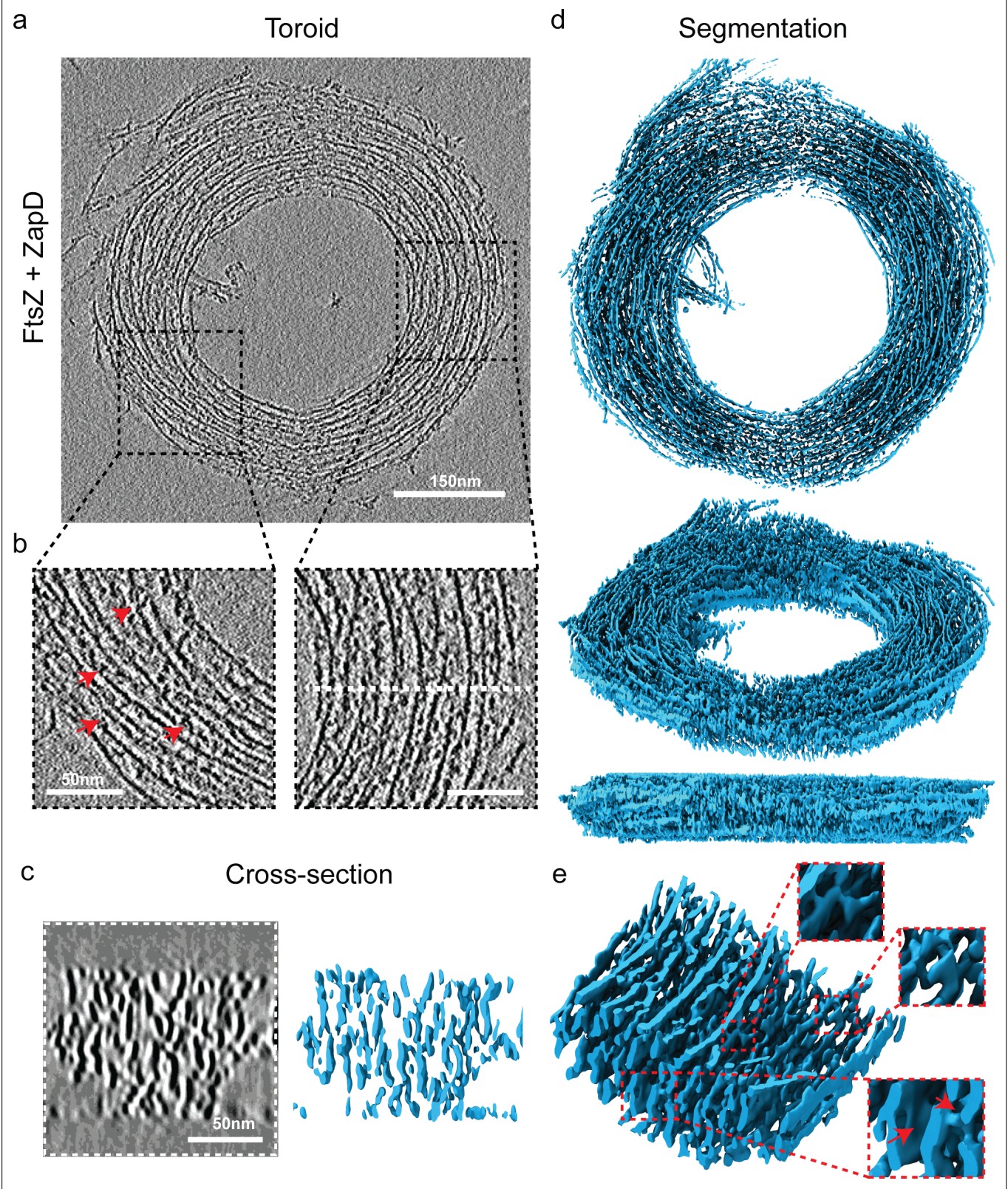

**Figure 3.** Three-dimensional (3D) structure of FtsZ toroids revealed by cryo-electron tomography (cryo-ET). (**a**) Representative tomographic slice of an FtsZ toroid resulting from the interaction of FtsZ with ZapD. The image is the average of five 0.86-nm-thick tomographic slices (total thickness of 4.31 nm) of the reconstructed tomogram around the equatorial plane of a single FtsZ toroid. The concentrations of FtsZ and ZapD were 10 µM, and 1 mM of GTP was added to initiate polymerization under working conditions (50 mM KCl, 50 mM Tris-Cl, 5 mM MgCl$_2$, pH 7). (**b**) Close-up views of the

*Figure 3 continued on next page*

*Figure 3 continued*

toroidal structure show the alignment of the FtsZ filaments forming the toroid. Red arrows indicate the presence of connections between filaments. (**c**) The tomographic slice in the XZ plane (left) shows the cross-section corresponding to the area marked by the white dotted line in **b** This image is the average of nine tomographic slices (total thickness of 7.74 nm) from the denoised tomogram. The isosurface of the cross-section (right) shows the vertical alignment and stacking of the FtsZ filaments within the toroid. This suggests that the interaction between FtsZ filaments and ZapD is mainly along the Z direction. FtsZ filaments are represented in blue. (**d**) Isosurface of the FtsZ toroid shown in **a**. It was extracted from the reconstruction of the denoised tomographic volume and positioned in different views to facilitate its visualization: (top) front view, (middle) side view, and (bottom) lateral view. The toroid has a diameter of ~552 nm and a height of ~92 nm. (**e**) A close-up view of the segmented toroidal structure. It shows the complex internal organization of filaments assembling the toroid. It corresponds to a zone within the toroid shown in **b** on the right. Close-up views of the isosurface show different connections between filaments. The segmentation shown has a width of 136 nm × 101 nm and a height of 64 nm.

The online version of this article includes the following figure supplement(s) for figure 3:

**Figure supplement 1.** FtsZ structures found at low and high ZapD concentrations.

**Figure supplement 2.** Segmentation of an FtsZ toroid imaged using cryo-electron tomography (cryo-ET).

**Figure supplement 3.** Cross-section of the toroid showing the elongated structures in the Z-axis.

**Figure supplement 4.** Segmentation of FtsZ filaments.

through ZapD proteins acting as bridging units, forming a complex 3D mesh (*Figure 4c and d* and *Video 2*). We also identified potential ZapD proteins that decorate individual FtsZ filaments and can link them to other nearby filaments (*Figure 4b*). Furthermore, some filaments exhibited multiple crosslinks, leading to stronger attachments between them (*Figure 4d*). Estimation of the precise number of ZapD molecules per FtsZ or the number of bonds per filament is challenging due to their inherently heterogeneous organization. However, we could observe a high number of connections stabilizing the toroidal structure. The short filament length, which results in gaps between adjacent filaments, does not correlate with a higher number of ZapD connections (*Figure 4—figure supplement 1*), indicating that ZapD is able to crosslink filaments in all directions without causing filament breakage. This could play an important mechanistic role in the functionality of the FtsZ macrostructures.

Cryo-ET imaging of ZapD-mediated FtsZ toroidal structures revealed a preferential vertical stacking and crosslinking of short FtsZ filaments, which are also crosslinked laterally and diagonally, allowing for filament curvature and resulting in a toroidal structure observed for the first time following the interaction between FtsZ and one of its natural partners in vitro.

## High concentrations of ZapD promote the structural reorganization of FtsZ polymers into straight bundles by tightly packed crosslinking

Having characterized the higher-order FtsZ structures promoted at equimolar ZapD binding densities, we addressed the effect of increasing the ZapD density on the structural organization of the ZapD-FtsZ polymers. At high concentrations of ZapD (typically 40–60 µM, representing a molar ratio of approximately one to four or six to FtsZ), we observed the formation of straight bundles with striated patterns between the FtsZ filaments (*Figure 5—figure supplement 1*), as well as the presence of some toroidal structures (*Figure 2—figure supplement 1a*). Here, the high concentration of ZapD molecules increased the number of links between the filaments and ultimately promoted the formation of straight bundles, indicating that the assembly of FtsZ-ZapD structures is a reversible process that strongly depends on the amount of ZapD proteins crosslinking the filaments. Toroids and curved bundles always coexist, but the predominance shifts from toroids to straight bundles at high ZapD concentrations. This shift suggests that the number of crosslinks

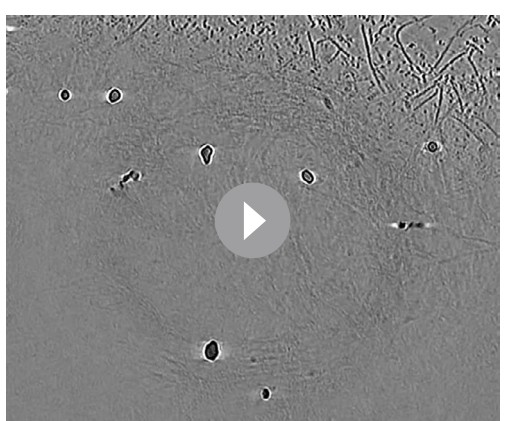

**Video 1.** Tomogram of an FtsZ toroidal structure promoted by ZapD shown in *Figure 3a*, followed by its segmentation. FtsZ filaments are in blue. Successive rotations of the segmented volume allow us to visualize the structure of the toroid in three dimensions (3D).
https://elifesciences.org/articles/95557/figures#video1

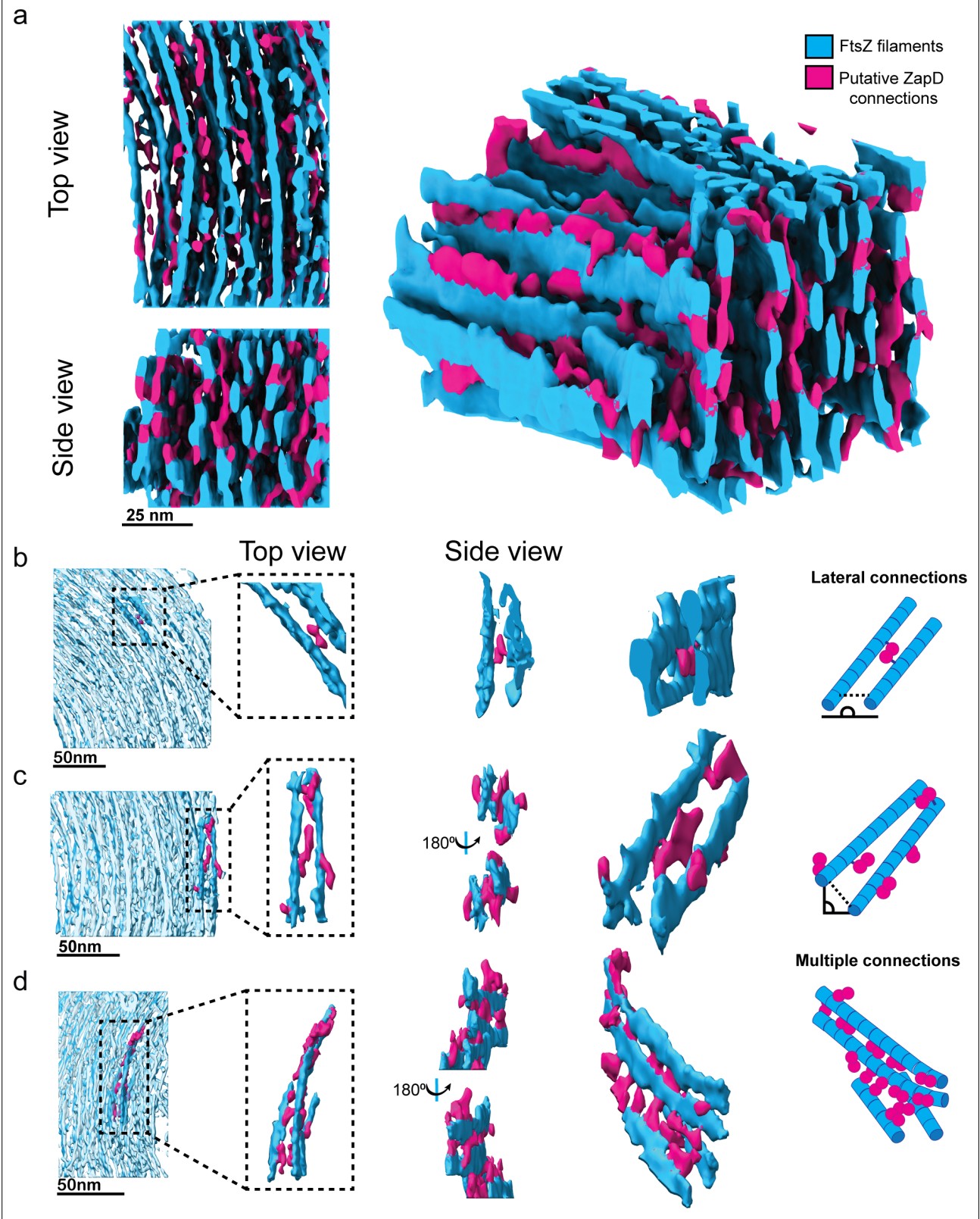

**Figure 4.** FtsZ filaments are connected by putative ZapD crosslinkers to assemble the toroidal structure. (**a**) Top (left, top), side (left, bottom), and lateral (right) views of the isosurface from a region within the toroidal structure shown in *Figure 3a*. The FtsZ filaments are colored in blue, while filament connections or putative ZapD proteins are labeled in magenta to facilitate interpretation of the results. Other putative ZapD proteins decorating the FtsZ filaments were not labeled in magenta because they were not forming any clear linkage between the filaments. The segmentation shown has a

*Figure 4 continued on next page*

*Figure 4 continued*

width of 73 nm × 101 nm and a height of 64 nm. (**b–d**) Various examples of filament connections by putative ZapD proteins within the toroid. Same color code as in (a). From left to right, the localization of the analyzed region, a close-up view of the structure of interest, different views of the crosslinkers, and a schematic illustrating the interpretation of the data. The schematic (right) shows the localization of ZapD proteins (magenta) and FtsZ filaments (blue). (**b**) Lateral connection of two FtsZ filaments by a putative ZapD dimer. In this example, the attachment of each globular density or putative ZapD monomer was bound to each filament, allowing for lateral binding. (**c**) Putative ZapD connections stabilizing two filaments by a lateral interaction. Additional ZapD decorations attached to only one of the filaments appear to be available for other filament connections. (**d**) Multiple ZapD proteins can connect to filaments and stabilize the interaction. First, the two upper filaments are connected vertically by several putative ZapDs. The lower filament connects vertically in an oblique angle to the nearest neighboring filament. In the upper part, additional decorations or putative ZapD proteins would be available to establish further interactions forming a three-dimensional (3D) mesh.

The online version of this article includes the following figure supplement(s) for figure 4:

**Figure supplement 1.** The FtsZ toroid is formed by short and discontinuous filaments crosslinked by ZapD.

**Figure supplement 2.** FtsZ single and double filaments found at different FtsZ and ZapD ratios.

between filaments can modulate the properties of the assembled higher-order structures, resulting in the reorganization of the toroidal structures into straight bundles upon ZapD saturation (***Figure 4— figure supplement 2***). We then explored how higher densities of ZapD lead to the formation of straight bundles using cryo-ET.

Using the same approach to visualize the toroids, we collected tomograms of the straight bundles and extracted their isosurfaces (***Figure 5a***, ***Figure 3—figure supplement 1b***, ***Figure 5—figure supplement 1***, and ***Video 3***). The straight bundles are formed at high ZapD concentrations and consist of a highly organized stack of well-aligned FtsZ filaments (***Figure 5b***). The connection of filaments by multiple putative ZapD proteins results in the straightening of the structure of the filaments (***Lu et al., 2000***; ***Erickson and Osawa, 2017***; ***Figure 5c***). Interestingly, ZapD crosslinks FtsZ filaments vertically, forming a row of ZapD proteins interacting with both filaments (***Figure 5d***). Here too, structural analysis by subtomogram averaging was not possible due to the crowding, preferential orientation, and heterogeneity of the connections. The distance between ZapD proteins provided a mean value of 4.5±0.5 nm (***Figure 5—figure supplement 2a***), consistent with the size of FtsZ proteins (***Huecas et al., 2007***). This observation suggests that most FtsZ proteins interact with the ZapD dimers that crosslink the filaments, in agreement with the ZapD enrichment in the bundles observed by sedimentation assays (***Figure 1—figure supplement 5***). In addition, we found that the presence of ZapD increased the distance between two FtsZ filaments connected by ZapD proteins compared with the spacing in the absence of ZapD, regardless of the amount of ZapD connections (from 5.9±0.8 to 7.88±2 nm in toroids and 7.6±1.5 nm in straight bundles; ***Figure 5—figure supplement 2b***). This indicates that ZapD not only connects neighboring FtsZ filaments but spreads them apart, which could have important implications for the functionality of the Z ring.

These results point out the relevant role of the number of connections between filaments (driven by ZapD binding) in the crosslinking mechanism (***Figure 6***). The bundles formed at the high ZapD concentrations tested (up to four molar excess of ZapD to FtsZ) correspond to a binding saturation regime in which a ZapD dimer binds two FtsZ molecules connecting two filaments, straightening their structure and arranging them into large, highly organized straight bundles. In contrast, fewer connections provide the freedom to form toroidal structures. This observation confirms that a certain number of ZapD-driven bonds are optimal for maintaining the preferential curvature of FtsZ filaments and allowing toroid assembly, establishing a structural dependence of assembled bundles on the number of ZapD-mediated bonds.

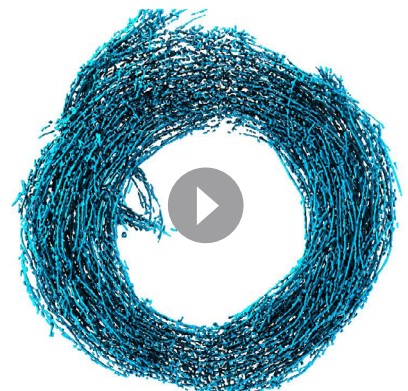

**Video 2.** Isosurface of the toroid shown in ***Figure 2a***. FtsZ filaments are shown in blue and putative ZapD connections in magenta. A close-up view and rotations of the segmented volume show the filament meshwork and the connections by ZapD in three dimensions (3D). https://elifesciences.org/articles/95557/figures#video2

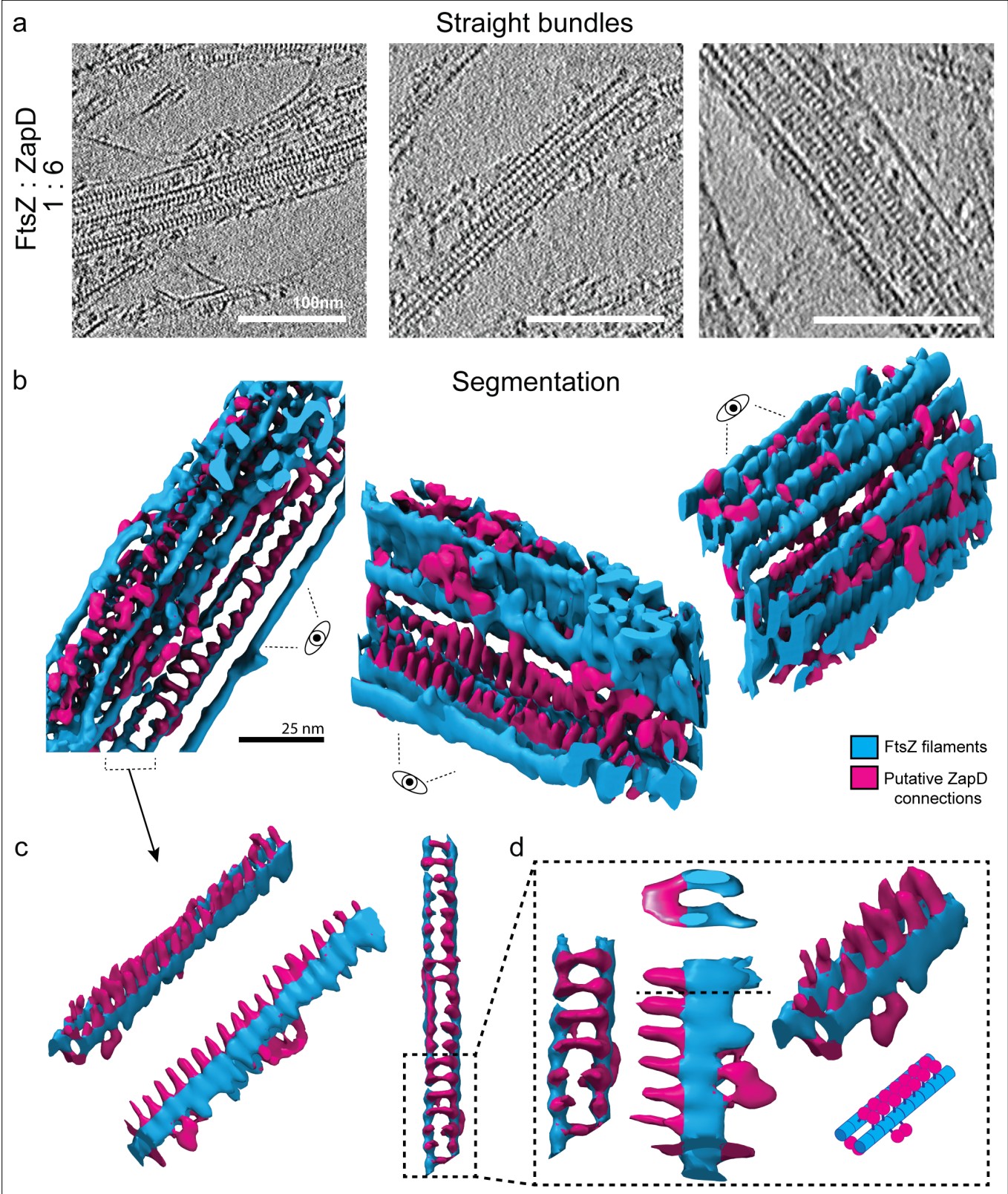

**Figure 5.** Formation of straight FtsZ bundles is driven by high ZapD crosslinking from above. (**a**) Representative tomographic slices of straight FtsZ bundles resulting from the interaction of FtsZ with high ZapD concentrations under working conditions (50 mM KCl, 50 mM Tris-Cl, 5 mM MgCl$_2$, pH 7). The concentrations of FtsZ and ZapD were 10 μM and 60 μM, respectively, and 1 mM of GTP was added to trigger polymerization. The straight bundles were found only at high ZapD concentrations. The image is the average of five 0.86-nm-thick tomographic slices (total thickness of 4.31 nm) of

*Figure 5 continued on next page*

*Figure 5 continued*

the reconstructed tomogram. Scale bars are 100 nm. (**b**) Isosurface of the straight bundles from the denoised tomographic volume. FtsZ filaments are colored in blue and putative ZapD connections in magenta. Three different views (top [left] and side views [middle, right]) are shown. Straight bundles are organized in a regular organization. Multiple bonds between filaments are formed vertically by putative ZapDs vertically crosslinking two FtsZ filaments with a regular spacing of 4.5±0.5 nm between ZapD dimers. In addition, lateral connections were also found, connecting pairs of stabilized filaments to each other and eventually assembling the straight bundle. (**c**) Different views of one of the isolated filaments from the straight bundle. A side view of the filaments (middle) shows a spike-like structure regularly located at the top of the FtsZ filaments, connecting them vertically as observed in the top view (right). (**d**) Different close-up views of the filament structure shown in (**c**). In the cross-section of the structure (middle, top), it is clearly visible that the ZapD proteins connect the two filaments vertically and from above, forming a bridge over them. A schematic of the proposed interaction (right, bottom) shows the position of putative ZapD dimers in this structure.

The online version of this article includes the following figure supplement(s) for figure 5:

**Figure supplement 1.** Segmentation of straight bundles imaged by cryo-electron tomography (cryo-ET).

**Figure supplement 2.** Distance between FtsZ filaments and ZapD-associated filaments.

**Figure supplement 3.** mZapD binds FtsZ-GDP and can promote bundling of FtsZ filaments.

**Figure supplement 4.** mZapD can bundle FtsZ without promoting toroids.

## Dimerization of ZapD is essential for the formation of organized higher-order FtsZ structures

Our results suggest that ZapD dimers connect two FtsZ filaments, although the role of dimerization has not yet been experimentally established (*Schumacher et al., 2017*). We therefore investigated whether the dimerization of ZapD was essential for assembling straight bundles and toroidal structures. To this end, we produced a ZapD mutant ('mZapD') by substituting three amino acids (R20, R116, and H140) involved in its dimerization by alanine, thus decreasing the stability of the dimerization interface. The dimerization of mZapD was tested by AUC analysis, which showed a percentage of monomeric protein that was not present in the wild-type ZapD (*Figure 5—figure supplement 3a*). mZapD can interact with unpolymerized FtsZ-GDP as evidenced by fluorescence anisotropy and FCS (*Figure 5—figure supplement 3b and c*). mZapD is still able to promote FtsZ bundle formation, showing an ~50% lower turbidity signal than for wild-type ZapD even at higher concentrations (*Figure 5—figure supplement 3d*). We also visualized these bundles by cryo-EM and observed the formation of thinner bundles (*Figure 5—figure supplement 4b and c*). However, toroidal structures and straight bundles were absent in these samples even at high concentrations of mZapD. This indicates that a stable dimerization interface is required to assemble complex ZapD-mediated structures (*Figure 5—figure supplement 4a*).

These results demonstrated that dimerization of ZapD is essential to promote the assembly of organized high-order FtsZ structures such as toroids and straight bundles, highlighting the structural importance of stable ZapD-driven connections between filaments.

## Discussion

In this study, we used biochemical reconstitution, cryo-EM, and cryo-ET to gain new insights into the structure and assembly of the bacterial divisome. Filament crosslinking by Zap proteins plays a crucial role in the assembly and stabilization of the Z ring. However, the underlying mechanisms of this process remain to be elucidated. To better understand these mechanisms, we investigated the interaction between FtsZ and ZapD, one of the stabilizers of the division ring.

Our results indicate that ZapD crosslinks FtsZ in solution to form 3D toroidal structures

**Video 3.** Tomogram of an FtsZ straight bundle formed at a high concentration of ZapD proteins shown in *Figure 5b*. Successive segmentation of the tomogram with FtsZ filaments labeled in blue and putative ZapD connections in magenta. Rotations and close-up views help the interpretation of the data and show the three-dimensional (3D) structure of the straight bundle.

https://elifesciences.org/articles/95557/figures#video3

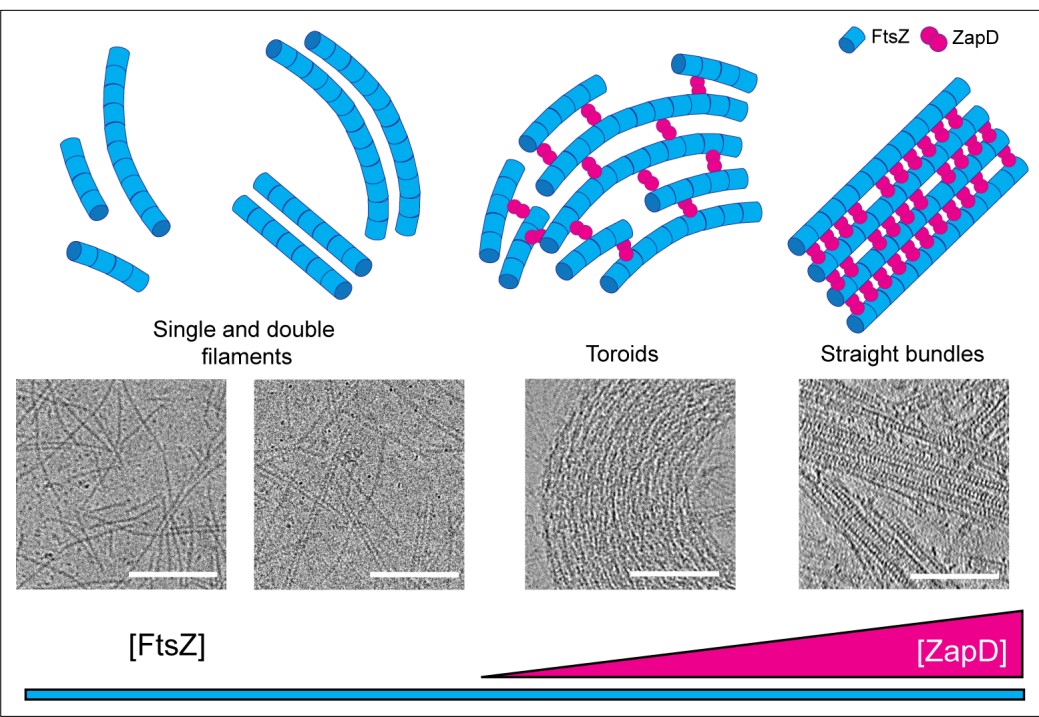

**Figure 6.** The amount of ZapD connections modulates the spatial organization of FtsZ filaments into higher-order structures. (Top) Simplified scheme of the higher-order FtsZ structures formed in the presence of increasing concentrations of ZapD. These schemes do not consider all the possible interactions between ZapD and FtsZ. (Bottom) Cryo-electron microscopy (cryo-EM) images of the structures shown in the schemes. In the absence of ZapD, FtsZ filaments can interact laterally to form double filaments upon GTP binding. At low concentrations of ZapD, only a few ZapD-mediated bonds are formed, favoring the formation of small, curved bundles. At equimolar concentrations of ZapD and FtsZ, more ZapD-mediated bonds are formed, particularly from above the filaments, but also laterally and diagonally, allowing filament curvature and favoring the assembly of toroidal structures. At saturating ZapD concentrations, the filaments are straightened up by regular ZapD crosslinking vertically, resulting in the formation of straight bundles. Overall, the assembly of higher-order FtsZ structures depends on the number of vertical crosslinks through ZapD dimers. Some intermediate states are expected between the structures shown. Scale bars are 100 nm.

characterized by short, discontinuous FtsZ filaments that are vertically stacked and crosslinked, as well as diagonally and laterally crosslinked, as revealed by cryo-ET. The relative concentrations of FtsZ and ZapD dictate the assembly geometry. Remarkably, FtsZ toroids emerge as the dominant structure when both proteins are present in equimolar ratios, whereas straight bundles are assembled at high concentrations of ZapD. Our study highlights the importance of the balance between vertical and lateral or diagonal crosslinking to allow for the filament curvature observed in the toroids, which is absent in the straight bundles where vertical crosslinking is dominant.

The structural characterization of the toroids by cryo-ET allowed us to resolve the overall structure and identify putative ZapD dimers that crosslink the filaments. However, we were unable to obtain detailed structural information about the ZapD connectors due to the heterogeneity and density of the toroidal structures, which showed significant variability in the conformations of the connections between the filaments in all directions. These results are consistent with the observation that ZapD interacts with FtsZ through its central hub, which provides additional spatial freedom to connect other filaments in different conformations. This flexibility allows different filament organizations and contributes to structural heterogeneity. In addition, these results suggest that these crosslinkers can act as modulators of the dynamics of the ring structure, spacing filaments apart and allowing them to slide in an organized manner (*Szwedziak et al., 2014*; *Nguyen et al., 2021*; *Lan et al., 2009*; *Hörger et al., 2008*). The ability of FtsZ to treadmill directionally (*Whitley et al., 2021*; *Loose and Mitchison, 2014*; *Ramirez-Diaz et al., 2018*; *Yang et al., 2017*; *Bisson-Filho et al., 2017*), together with the parallel or antiparallel arrangement of short,

transiently crosslinked filaments, is considered essential for the functionality of the Z ring and its ability to exert constrictive force (*Whitley et al., 2021*; *Erickson et al., 2010*; *Nguyen et al., 2021*; *Mateos-Gil et al., 2019*; *Erickson and Osawa, 2017*). Thus, Zap proteins can play a critical role in ensuring correct filament placement and stabilization, which is consistent with the toroidal structure formed by ZapD.

Our observation of toroidal FtsZ structures promoted by ZapD in solution is consistent with the observation of pre-curved FtsZ protofilaments in circular assemblies attached to model membranes in various in vitro systems (*Loose and Mitchison, 2014*; *Ramirez-Diaz et al., 2018*; *Vélez, 2022*). Additionally, FtsZ can form ring-shaped structures with diameters ranging from 100 to 200 nm on surfaces, becoming more pronounced when adsorbed to lipid, carbon, or mica surfaces (*Huecas et al., 2007*; *Mingorance et al., 2005*; *González et al., 2005*), or in the presence of molecular crowders like methylcellulose (*Popp et al., 2009*). Cryo-EM of concentrated (50 µM) FtsZ from *B. subtilis* with GMPCPP revealed that these protofilaments often coalesce into spirals or toroids, forming large aggregates (*Huecas et al., 2017*). These toroidal structures correspond to the curved conformations of FtsZ polymers observed in various bacterial species, which are thought to contribute to the generation of constriction forces (*Erickson and Osawa, 2017*). In *E. coli,* approximately 5000 FtsZ monomers (about one third of the intracellular concentration of FtsZ) could circle the cell six to eight times (*Stricker et al., 2002*), suggesting a discontinuous toroidal assembly. ZapD, present at approximately 500 molecules per cell (*Durand-Heredia et al., 2012*), represents only a small fraction of the FtsZ molecules and favors the formation of toroidal structures in the absence of interactions with other FtsZ-associated proteins. Higher concentrations of ZapD would be required to form straight bundles.

The persistence length and curvature of FtsZ filaments are optimized to form bacterial-sized ring structures (*McQuillen and Xiao, 2020*; *Erickson and Osawa, 2017*). ZapD helps to stabilize these toroids by crosslinking and increasing the spacing between FtsZ filaments from 5.9±0.8 to 7.9±2 nm in toroids and 7.6±1.5 nm in straight bundles (*Figure 5—figure supplement 2b*). This increased spacing promotes a more dynamic organization, providing functional flexibility in bacteria. The linker connecting ZapD to FtsZ can also modulate filament spacing and curvature (*Huecas et al., 2017*; *Sundararajan and Goley, 2017*; *Sundararajan et al., 2018*). We observe a mixture of curvatures in the internal organization of the toroids. While FtsZ filaments have a preferred curvature, the density of ZapD connections forces the structure to adapt and align with neighboring filaments, thereby affecting FtsZ curvature. However, the precise molecular mechanism linking ZapD binding and polymer curvature remains to be elucidated.

One of the most significant findings of this study is that the amount of ZapD bound to FtsZ filaments greatly influences the structural organization of the resulting higher-order polymer. At lower concentrations of ZapD, which are necessary for bundling (equimolar mixtures of ZapD and FtsZ), toroids are the most prominent structures, with a binding stoichiometry of ZapD in the polymer of 0.3–0.4 moles of ZapD per mole of FtsZ (equivalent to one ZapD dimer for every four to six FtsZ molecules). As the concentration of ZapD increases, the binding stoichiometry of ZapD in the polymers increases to 1.1 moles of ZapD per mole of FtsZ (one ZapD dimer for every two FtsZ molecules). This leads to a reorganization of the polymer bundles, resulting in straight structures that predominantly crosslink the FtsZ filaments, a configuration not seen at lower ZapD concentrations (*Figure 5*). The increase in the number of ZapD-FtsZ contacts likely reduces the flexibility needed to form toroidal structures, compelling the filaments to adopt a larger, straight bundle conformation. We observed slight differences in the spacing between filaments (*Figure 5—figure supplement 2b*), with a broader range of distances between filaments in the toroids compared to the straight bundles. Thus, ZapD contributes to the variability of spacing based on the number of connections formed.

We therefore hypothesize that the assembly of functional, curved FtsZ macrostructures occurs only within a specific stoichiometric range of ZapD-FtsZ interactions. An increase in crosslinking at higher ZapD concentrations appears to cause the FtsZ polymer to form rigid, straight bundles (*Figure 6*). Additionally, only a robust ZapD dimer can form FtsZ toroids and straight filaments, indicating that a certain binding strength is necessary to bundle the filaments and maintain their structure. These observations, along with the high conformational variability found in ZapD connections, suggest that ZapD can modulate the behavior of the entire structure through a concentration-dependent mechanism. This structural reorganization highlights the polymorphic nature of the assemblies formed by FtsZ (*Mateos-Gil et al., 2019*; *Monterroso et al., 2013*).

We also demonstrated that ZapD-mediated FtsZ toroids replicate some of the structural features of the Z ring architecture observed in vivo. Based on super-resolution imaging and cryo-ET, the current model of the Z ring describes a somewhat discontinuous and heterogeneous structure composed of randomly overlapping filaments (*Szwedziak et al., 2014*; *Fu et al., 2010*; *Rowlett and Margolin, 2014*). These filaments arrange into a belt-like macromolecular entity, containing nodes with a higher density of dispersed filaments confined to a toroidal zone approximately 80–100 nm wide (toroidal height) and about 40–60 nm thick, located 13–16 nm below the inner membrane (*Szwedziak et al., 2014*; *Buss et al., 2013*; *Fu et al., 2010*; *Lyu et al., 2016*; *Rowlett and Margolin, 2014*; *Holden et al., 2014*; *Szwedziak and Ghosal, 2017*). The toroidal structures promoted by ZapD exhibit dimensions within this range. However, their thickness and multilayering are significantly greater, likely due to the absence of other cellular components competing for binding with FtsZ. The Z ring model also suggests that FtsZ filaments are weakly associated with one another through protein factors such as Zaps and weak FtsZ-FtsZ interactions, which may be crucial for their functionality (*McQuillen and Xiao, 2020*; *Huecas et al., 2017*; *Sundararajan and Goley, 2017*; *Haeusser et al., 2015*). Our findings indicate that one of the Zap proteins can stabilize and enhance associations between neighboring FtsZ filaments. Studying macrostructures stabilized by natural crosslinkers is vital for understanding the molecular function of the division machinery.

We believe that the intrinsic features of the toroidal structures share commonalities with the bacterial division ring. Despite the differences and limitations that may arise from an in vitro approach, the structures observed following ZapD crosslinking of FtsZ filaments may reveal inherent features that occur in vivo. The current model of the division ring consists of an array of filaments loosely connected by crosslinkers at the center of the cell, forming a ring. This model is consistent with our findings, although many questions remain regarding the structural organization of the Z ring within the cell. ZapD binds to FtsZ vertically, allowing either ZapD or FtsZ to interact with the plasma membrane. In straight bundles, this facilitates the stacking of straight FtsZ filaments, while for toroids, ZapD can also bind FtsZ filaments diagonally. This less compact arrangement could allow bending of the FtsZ filaments and adjustment of the toroid size.

The binding of ZapA, a key member of the FtsZ-associated proteins family, to FtsZ in equimolar mixtures induces the effective alignment and straightening of FtsZ filaments tethered to lipid bilayers by FtsA protein, as demonstrated through high-resolution fluorescence microscopy and precise quantitative image analysis (*Caldas et al., 2019*). This interaction reveals a structure comprising one ZapA tetramer for every four FtsZ molecules in the polymer, highlighting the organized nature of these interactions. It is noteworthy that these straight, parallel polymers – manifested when ZapA occupies all binding sites – bear striking resemblance to the straight bundles identified in this study at high concentrations of ZapD, indicative of binding saturation. Understanding how the associative states of ZapA (as tetramers) and ZapD (as dimers), together with membrane tethering, influence the predominant structures formed in both systems is essential. The complexity of the division system raises important questions about the interaction dynamics between FtsZ and the plasma membrane. The competitive nature of the division components to engage with FtsZ and modulate its functionality remains to be thoroughly elucidated. It is important to note that FtsA and ZipA have a greater affinity for the CCTP of FtsZ than ZapD. Our cryo-ET data on straight bundles provide new perspectives on how ZapD-FtsZ structures can effectively bind to the plasma membrane; in particular, the CCTP of parallel FtsZ filaments is oriented upward, allowing direct membrane binding or interaction with ZapDs that reinforce these filaments from above, rather than from the side, as previously suggested (*Schumacher et al., 2017*).

In conclusion, this study extends the understanding of the intricate structural organization and filament crosslinking of FtsZ polymers facilitated by ZapD, which is critical for the formation and maintenance of the division ring. The revelation that ring morphology depends on the binding stoichiometry between FtsZ and the crosslinking ZapD is invaluable, particularly as we integrate this knowledge into current reconstitution assays in membrane systems. This knowledge is central to unraveling the role of FtsZ-associated proteins in coordinating and maintaining division ring formation. In addition, our findings could greatly enhance the understanding of how polymeric cytoskeletal networks are remodeled during essential cellular processes such as cell motility and morphogenesis. Although conventional wisdom points to molecular motors as the primary drivers of filament remodeling through energy consumption, there is increasing evidence that there are alternative mechanisms that do not rely on

such energy, instead harnessing entropic forces via diffusible crosslinkers (*Braun et al., 2016*). This approach may also be applicable to ZapD and FtsZ polymers, suggesting a promising avenue for optimizing conditions in the reverse engineering of the division ring to enhance force generation in minimally reconstituted systems aimed at achieving autonomous cell division.

## Methods

### Protein purification

ZapD protein has been overproduced and purified following the procedure previously described in *Durand-Heredia et al., 2012*, with some modifications. The bacterial strain was an *E. coli* BL21 (DE3) transformed with pET28b-h10-smt3-ZapD fusion plasmid (*Durand-Heredia et al., 2012*). The cells were sonicated for 6–8 cycles of 20 s and centrifuged at 40,000 rpm in MLA 80 rotor (Beckman Coulter) for 45 min. The protein was eluted by chromatography in 5 mL HisTrap (GE Healthcare) and digested with Smt3-specific protease Ulp1 (SUMO protease encoded in Rosetta pTB145) at 1:1 molar relation. Digestion proceeded for two dialysis sessions of 1 hr at 4°C to avoid protein precipitation. ZapD and His-Smt3 were separated by chromatography in 5 mL HisTrap (GE Healthcare), then the protein was eluted in a 5 mL HiTrap Desalting column (Healthcare) to eliminate completely possible traces of free phosphates in the buffer. Final concentration of ZapD was determined by absorbance at 280 nm using an extinction coefficient ($\varepsilon$) of 25230 M$^{-1}$ cm$^{-1}$, and ZapD purity was checked by SDS-PAGE. Protein fractions were frozen in buffer 50 mM KCl, 50 mM Tris-Cl, 10 mM MgCl$_2$, 0.2 mM DTT, and 2% glycerol. ZapD mutant (mZapD) was purified following the same procedure using the corresponding plasmid. *E. coli* FtsZ was overproduced and purified following the calcium precipitation method described previously (*Rivas et al., 2000*). Briefly, *E. coli* BL21 (DE3) pLysS cells were transformed with pET11b-FtsZ, grown in LB medium, and selected with Ampicillin 100 µg/mL. After induction and growth, the pellet was resuspended in PEM buffer (50 mM PIPES-NaOH, pH 6.5, 5 mM MgCl$_2$, 1 mM EDTA) and disrupted using a tip sonicator for 3–4 cycles. The lysate was then separated by centrifugation for 30 min at 20,000 × *g* at 4°C, and the supernatant was mixed with 1 mM GTP, 20 mM CaCl$_2$, and incubated at 30°C for 15 min to induce FtsZ polymerization and bundling. Subsequently, the FtsZ bundles were pelleted by centrifugation for 15 min at 20,000 × *g* at 4°C, and the pellet was resuspended in PEM buffer and centrifuged again for 15 min at 20,000 × *g*, 4°C, collecting the supernatant. Precipitation and resuspension steps were repeated to improve the purification. The buffer was then exchanged using an Amicon Ultra-0.5 centrifugal filter unit 50 kDa (Merck KGaA). FtsZ purity was checked by SDS-PAGE, and concentration was determined by absorbance at 280 nm using an extinction coefficient ($\varepsilon$) of 14000 M$^{-1}$ cm$^{-1}$ (*Rivas et al., 2000*). Protein solutions were aliquoted, frozen in liquid nitrogen, and stored at –80°C until further use.

### Plasmid design and molecular cloning

To generate the ZapD mutant (mZapD), seamless cloning method was used according to the provider's protocol (Thermo Fisher Scientific/Invitrogen GeneArt Seamless Cloning and Assembly Enzyme Mix [A14606]) using the plasmid pET28b-h10-smt3-ZapD and the following primers (R20A) 'GAAA AAATGCGTACATGGCTGGCTATTGAGTTTT', 'GGCAATTTCTGCGTGAAGAT**GCT**TTGATTGCTC', and 'CTTTGATTTACCTACATTG**GCT**ATTTGGCTGC' to substitute the amino acids 20 (R), 116 (R), and 240 (H) for A. All enzymes for cloning were from Thermo Fisher Scientific (Waltham, MA, USA). Briefly, DNA fragments were amplified with Phusion High-Fidelity DNA Polymerase and origo primers (Sigma-Aldrich, St. Louis, MO, USA). Then, PCR products were treated with DpnI and combined using GeneArt Seamless Cloning and Assembly Enzyme Mix. The plasmid was propagated in *E. coli* OneShot TOP10 (Thermo Fisher Scientific) and purified using NucleoBond Xtra Midi kit (Macherey-Nagel GmbH, Düren, Germany). Directed site mutagenesis was made by substituting three amino acids in the ZapD sequence by Alanine (R20, R116, H140). The plasmid was then verified using Sanger Sequencing Service (Microsynth AG, Balgach, Switzerland).

### Protein labeling

The protein was covalently labeled at the amine groups with Alexa Fluor 488 carboxylic acid succinimidyl ester, a fluorescent dye (Molecular Probes/Invitrogen), under conditions that ensured minimal interference of the dye with FtsZ assembly, as described in *Reija et al., 2011*. ZapD and mZapD

were labeled with ATTO-647N carboxylic acid succinimidyl ester dye in the amino group. Before the reaction, ZapD was dialyzed in 20 mM HEPES, 50 mM KCl, 5 mM MgCl$_2$, pH 7.5, and the probe was dissolved in dimethylsulfoxide. The reaction was allowed to proceed for 35–60 min at room temperature (RT) and stopped with 10% Tris-HCl 1 M. The free dye was separated from labeled protein by a Hi-TRAP Desalting column (GE Healthcare). The final degree of labeling of FtsZ and ZapD was calculated from the molar absorption coefficients of the protein and the dye. It was around 0.5 moles of probe per mole of FtsZ and around 0.3/0.4 moles of dye per mole of ZapD.

## AUC, sedimentation velocity, and SE

Sedimentation velocity assays were performed to detect the homogeneity and association state of individual proteins and the stoichiometry of the formed protein-protein complexes. In brief, the experiments were carried out at 43–48 K rpm in an Optima XL-I analytical ultracentrifuge, equipped with UV-Vis absorbance and Rayleigh interference detection systems. The sedimentation coefficient distributions were calculated by least-squares boundary modeling of sedimentation velocity data using the *c*(*s*) method (*Schuck, 2000*) as implemented in the SEDFIT program. The s-values of the present species were corrected to standard conditions (pure water at 20°C, and extrapolated to infinite dilution) to obtain the corresponding standard s-values (s20,w) using the program SEDNTERP (*Laue et al., 1992*). Multi-signal sedimentation velocity data were globally analyzed by SEDPHAT software (*Schuck, 2003*) using the 'multi-wavelength discrete/continuous distribution analysis' model, to determine the spectral and diffusion deconvoluted sedimentation coefficient distributions, *ck*(*s*), from which the stoichiometry of protein complexes can be derived (*Balbo et al., 2005*). Sedimentation equilibrium (SE) of ZapD was carried out to confirm the association state of the protein in the same experimental conditions and concentration range tested by sedimentation velocity (2–30 µM). Short columns (100 µL) SE experiments were carried out at 14,000 and 10,000 rpm. Weight-average buoyant molecular weights were obtained by fitting a single-species model to the experimental data using the HeteroAnalysis program (*Cole, 2004*), once corrected for temperature and buffer composition with the program SEDNTERP (*Laue et al., 1992*).

## Turbidity assay

Turbidity of protein samples was collected by measuring the absorbance at 350 nm in a TECAN plate reader (Tecan Group Ltd., Mannedorf, Switzerland). All samples reached a final volume of 50 µL in a 364-Well Flat-Bottom Microplate (UV-Star, Greiner Bio-One GmbH) before measuring the absorbance. Different concentrations of FtsZ and ZapD were premixed in the well plate and measured before addition of GTP to extract subsequently the individual blank values. Concentrations of FtsZ and ZapD varied from 0 to 80 µM, and buffer conditions are specified in each graph and the caption of the figures ranging from 50 to 500 mM KCl, 6–8 pH, always supplemented with 50 mM Tris-Cl and 5 mM MgCl$_2$. Manual mixing was performed after addition of GTP, and orbital oscillations for 5 s in the TECAN were made prior to any measurement to avoid sedimentation of the samples. FtsZ and ZapD alone do not generate significant differences with the blank. Time measurements were taken as specified in each condition. Reported values are the average of 3–12 independent measurements ± standard deviation.

## GTPase activity of FtsZ

GTPase activity of FtsZ was measured by quantification of the inorganic phosphate with a colorimetric assay (BIOMOL GREENÒ kit from ENZO Life Sciences) for 2 min. 5 µM FtsZ was used in our standard buffer (5 mM MgCl$_2$, 50 mM Tris-HCl, 50 mM KCl, pH 7) or buffers at higher KCl concentrations (50–500 mM KCl) and polymerization was triggered by 1 mM GTP. ZapD was added at different concentrations and premixed with FtsZ before addition of GTP. 13 µL fractions were added to a 96-Well Flat-Bottom Microplate (UV-Star, Greiner Bio-One GmbH) every 20 s after addition of GTP and mixed with the Reaction buffer reaching 50 and 100 µL of BIOMOL GREEN reagent, to stop the reaction. After stopping the reaction, samples were incubated for 10 min at RT, and the absorbance was measured at 620 nm in a Varioskan Flash plate reader (Thermo Fisher Scientific, MA, USA). Concentrations of inorganic phosphate were calculated from a phosphate standard curve, while the GTPase activity reaction rate (V, mol P/mol FtsZ/min) was determined from the slope of the linear part of phosphate accumulation curves.

## FtsZ sedimentation assay

Purified ZapD (1–30 µM) was added to purified FtsZ (5 µM) in the working buffer (50 mM KCl, 50 mM Tris-HCl, 5 mM MgCl$_2$), and GDP or GTP (1 mM) was added last to trigger FtsZ polymerization. The reaction mixtures with a final volume of 100 µL were processed at RT and centrifuged at low speed (10,000 rcf) using a tabletop centrifuge. At that point, 90 µL of supernatant was carefully collected and loaded in clean tubes with 1x loading dye. The rest of the supernatant was discarded, and the pellets were resuspended in the original reaction volume buffer plus 1× loading dye (final concentration). The supernatants (10 µL) and pellets (10 µL) were resolved in a SDS-PAGE gel. The amount of FtsZ was estimated by image analysis using ImageJ.

## Preparation of EM grids

Cryo-EM grids were plunge-frozen with a Vitrobot Mk.IV (Thermo Fisher Scientific) using 3 µL of the samples applied to previously glow-discharged R 2/1 Holey Carbon Cu 200 mesh EM grids (Quantifoil). Samples were 10 µM FtsZ with or without ZapD or mZapD at different concentrations specified in each case (0–60 µM). Proteins were mixed in our working buffer containing 50 mM, 5 mM MgCl$_2$, 50 mM Tris-HCl, pH 7. Samples were incubated for 2 min after the addition of 1 mM GTP to trigger polymerization. The Vitrobot was set to 4°C, 100% humidity, blot time 3 s, blot force 3. Whatman no. 1 filter paper was used for blotting, and liquid ethane kept at liquid nitrogen temperatures was used as a cryogen for vitrification.

## Cryo-EM and cryo-ET

Cryo-EM/ET data were acquired on two microscopes as follows. Cryo-EM micrographs were acquired within SerialEM (*Mastronarde, 2005*) on a Talos Arctica transmission electron microscope (Thermo Fisher Scientific) operated at 200 kV, equipped with a Falcon III (Thermo Fisher Scientific) direct electron detector operated in integrating mode. Images were recorded at ×73,000 magnification (pixel size 1.997 Å) and ×92,000 magnification (pixel size 1.612 Å) at –2.5 to –5 µm target defocus with an approximate total electron dose of 60 electrons/Å².

Cryo-EM micrographs and cryo-ET tilt series were acquired with SerialEM on a Titan Krios G2 transmission electron microscope (Thermo Fisher Scientific) operated at 300 kV, equipped with a FEG, post-column energy filter (Gatan), and a K3 camera (Gatan) operated in electron counting mode. Micrographs were recorded at ×42,000 magnification (pixel size 2.154 Å) at –5 µm target defocus with an approximate total electron dose of 60 electrons/Å². Tilt series were recorded at ×42,000 magnification (pixel size 2.154 Å) at –5 µm target defocus with an approximate total electron dose of 100–140 electrons/Å². Acquisition was performed using a dose-symmetric tilt scheme, a tilt range of ±60°, an angular increment of 2°.

## Tomogram reconstruction

Tilt series preprocessing was performed using the TOMOMAN package (https://github.com/wan-lab-vanderbilt/TOMOMAN, *Wan, 2024*). In brief, MotionCor2 (*Zheng et al., 2017*) was used to align individual frames, followed by cumulative dose weighting using an exposure-dependent attenuation function, as described in *Schur et al., 2016*. Dose-weighted tilt series were aligned using IMOD (*Kremer et al., 1996*) either by employing gold fiducials (if available) or by patch tracking, and binned tomograms (pixel size 8.616 Å) were generated using weighted back projection. Stacks were split into odd and even frames to generate half-set tomograms which were used for training and subsequent denoising in cryoCARE (*Buchholz et al., 2019*).

Denoised tomograms were used directly for segmentation of the toroids, straight bundles, and individual filaments by cropping an area of interest and displaying the volume as an isosurface in UCSF ChimeraX (*Goddard et al., 2018*). Thresholds for volume extraction were 80–100 and the color used was #00AEEF (light blue) for FtsZ and #EC008C (magenta) for ZapD isosurfaces. Small particles were removed by using the 'dust' function with a size of 10. Different perspectives and zooms of the structures were manually adjusted for each figure in ChimeraX. To highlight connections between filaments, the corresponding parts with putative ZapD connections were manually cropped from the volume using the Volume Eraser tool in UCSF Chimera (*Pettersen et al., 2004*). Both cropped and original isosurfaces were superimposed to show the colored putative ZapDs in the structure. The missing wedge effect induces an elongation by a factor of 2 along the Z-axis. This elongation is

observed in the filaments of the 1:0 ratio toroids, whereas the elongation observed in the filaments of the 1:1 and 1:6 ratio toroids exceeds the missing wedge.

All micrographs and slices through tomograms were visualized using IMOD. Isosurface renderings of toroids, straight bundles, and individual filaments were displayed using UCSF ChimeraX and UCSF Chimera.

### Fluorescence anisotropy

Anisotropy measurements were performed using a TECAN plate reader (Tecan Group Ltd., Mannedorf, Switzerland). Excitation and emission wavelengths were 625 and 680 nm, respectively. ZapD or mZapD labeled with ATTO 647 N were used as fluorescence tracer with a final concentration of 150 nM of ATTO-647N and supplemented with unlabeled ZapD reaching a concentration of 5 µM. FtsZ was added at increasing concentrations to analyze their interaction. Binding affinities (apparent $K_d$) were determined by fitting the Hill equation to the normalized anisotropy data. Each condition was measured in three independent samples.

### Fluorescence correlation spectroscopy

FCS measurements were performed using a PicoQuant MicroTime200 system equipped with an Olympus 60x, NA1.2 UPlanApo water immersion objective. A pulsed 636 nm diode laser was used to excite fluorescence of Atto647N-labeled ZapD, mZapD, or free Atto647N carboxylate (for calibration). Three measurements of 60 s each were performed per sample at RT (21°C), and three samples per condition were measured. The repetition rate of the pulsed laser was 26.7 MHz, and the average power was 1 µW at the objective back pupil. Fluorescence was collected through the same objective, spatially filtered through a 50 µm diameter pinhole, and spectrally filtered through a 690/70 nm bandpass filter before being split by a neutral beam splitter onto two avalanche photodiodes (Excelitas SPCM-AQRH-14-TR). Photon count events were digitized using a PicoQuant TimeHarp 260 Nano TCSPC card. Time-correlated single photon counting information was used for monitoring data quality during measurements, but not in further analysis. Data was subjected to a burst removal algorithm similar to *Margineanu et al., 2007*, and only 'non-burst' data was used in correlation analysis to obtain statistics representative of the large number of small oligomer particles, ignoring rare large ones. Cross-correlation functions of the two detectors were calculated and fitted with a standard model for 3D diffusion in with rapid dye blinking:

$$G\left(\tau\right) = G_0 \left[ \frac{1 - F_B + F_B e^{\frac{-\tau}{\tau_B}}}{1 - F_B} \right] \frac{1}{1 + \frac{\tau}{\tau_d}} \sqrt{\frac{1}{1 + \frac{\tau}{S^2 \tau_d}}}$$

With amplitude $G_0$, diffusion time $\tau_d$, point spread function aspect ratio $S$, and blinking parameters $F_B$ and $\tau_B$. Custom software was written in Python for burst removal, correlation, and fitting, based on tttrlib 0.0.19 (https://github.com/Fluorescence-Tools/tttrlib, *Peulen and Hemmen, 2024*). The software is in ongoing development and available upon request.

### Image analysis

Electron microscopy images were processed and analyzed using IMOD and ImageJ. The dimensions of the toroidal structures, FtsZ bundles, and distances between filaments were manually measured using ImageJ and IMOD. Distances were plotted as histograms using Origin (OriginPro, Version 2019b. OriginLab Corporation, Northampton, MA, USA). For each toroid analyzed (*n*=67), the inner and outer diameters were measured by collecting the major and minor distances in each case. The circularity of the toroid was the result of the division between the minor diameter divided by major diameter for each toroid. The height of the FtsZ toroid was manually measured from the tomograms, collecting four measurements per toroid to assure a correct representation of the size (*n*=17). For the spacing between filaments, the space in between the filaments was measured manually in the XY plane by using IMOD. Each measurement represents only one FtsZ double filament or one FtsZ-ZapD double filament; however, the same bundle could be measured more than once as they are composed of multiple filaments. The measurements were collected from >3 independent samples. The distances between ZapDs connecting two FtsZ filaments were measured following the same methodology. The

mean value and standard deviation of different datasets were calculated and added to the figures together with the *n* used for each case.

## Acknowledgements

We thank Daniel Bollschweiller (MPIB cryo-EM facility) and Rafael Nuñez (CIB-CSIC imaging facility) for their help with the cryo-EM experiments. We also thank the MPIB core facility for assistance in protein purification, Michaela Schaper for plasmid cloning, Sigrid Bauer for lipid preparation, and Noelia Ropero, Katharina Nakel, and Kerstin Andersson for protein purification. We are also grateful to Miguel Robles-Ramos, Ana Raso, Hiromune Eto, Cristina Capitanio, Pedro Weickert, Kareem Al-Nahas, Yusuf Qutbuddin, Silvia Zorrilla, Henri Franquelim, Diego Ramirez, and Daniela Garcia-Soriano for helpful discussions and fruitful input in this work. This work was funded by the Max Planck-Bristol Centre for Minimal Biology (AM-S), the Deutsche Forschungsgemeinschaft (PS), the Deutsche Forschungsgemeinschaft (DFG) call ANR-DFG 2020 NLE through the grant JA-3038/2-1 (M Ja), the Germany's Excellence Strategy – EXC-2094 – 390783311 (J-HK) and the Spanish Government through Grants PID2019_104544GB-100 and PID2022_136951NB-100 (GR). MS-S was supported by the European Social Fund through Grant PTA2020-018219-I. JRL-O and MS-S acknowledge support from the Molecular Interactions Facility at the CIB Margarita Salas-CSIC. AM-S and J-HK are part of IMPRS-LS, and J-HK is also a CeNS Center for NanoScience associate.

## Additional information

### Funding

| Funder | Grant reference number | Author |
| --- | --- | --- |
| Max Planck-Bristol Centre for Minimal Biology | | Adrián Merino-Salomón |
| Deutsche Forschungsgemeinschaft | | Petra Schwille |
| Deutsche Forschungsgemeinschaft | JA-3038/2-1 | Marion Jasnin |
| Spanish National Plan for Scientific and Technical Research and Innovation | PID2019_104544GB-100 | Germán Rivas |
| Spanish National Plan for Scientific and Technical Research and Innovation | PID2022_136951NB-100 | Germán Rivas |
| European Social Fund Plus | PTA2020-018219-I | Mercedes Jiménez |
| Germany's Excellence Strategy | EXC-2094 – 390783311 | Jan-Hagen Krohn |

The funders had no role in study design, data collection and interpretation, or the decision to submit the work for publication. Open access funding provided by Max Planck Society.

### Author contributions

Adrián Merino-Salomón, Conceptualization, Data curation, Formal analysis, Validation, Investigation, Visualization, Methodology, Writing – original draft, Project administration, Writing – review and editing; Jonathan Schneider, Data curation, Investigation; Leon Babl, Tillman Schäfer, Data curation, Investigation, Methodology; Jan-Hagen Krohn, Data curation, Validation, Investigation; Marta Sobrinos-Sanguino, Juan Ramon Luque-Ortega, Data curation, Validation, Investigation, Methodology; Carlos Alfonso, Data curation, Formal analysis, Validation, Investigation, Methodology; Mercedes Jiménez, Conceptualization, Data curation, Supervision, Validation, Investigation, Methodology, Project administration; Marion Jasnin, Data curation, Formal analysis, Supervision, Validation, Investigation, Visualization, Writing – original draft, Project administration, Writing – review and

editing; Petra Schwille, Conceptualization, Resources, Supervision, Funding acquisition, Validation, Project administration, Writing – review and editing; Germán Rivas, Conceptualization, Resources, Formal analysis, Supervision, Funding acquisition, Validation, Visualization, Writing – original draft, Project administration, Writing – review and editing

## Author ORCIDs

Adrián Merino-Salomón ⬤ https://orcid.org/0000-0002-6132-5314
Jan-Hagen Krohn ⬤ https://orcid.org/0000-0002-7383-3535
Tillman Schäfer ⬤ https://orcid.org/0000-0002-9992-2501
Marion Jasnin ⬤ https://orcid.org/0000-0003-1726-4566
Petra Schwille ⬤ https://orcid.org/0000-0002-6106-4847
Germán Rivas ⬤ https://orcid.org/0000-0003-3450-7478

Reviewer #1 (Public review): https://doi.org/10.7554/eLife.95557.4.sa1
Reviewer #3 (Public review): https://doi.org/10.7554/eLife.95557.4.sa2
Author response https://doi.org/10.7554/eLife.95557.4.sa3

## Additional files

### Supplementary files

MDAR checklist

Source data 1. Raw data used to make the graphs shown in figures.

### Data availability

The data sets for all experimental conditions and graphs generated in this study are shown in the manuscript or provided in the supplementary materials/source data file.

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
