## [Editor Report · eLife Assessment]

The formation of the Z-ring at the time of bacterial cell division interests researchers working towards understanding cell division across all domains of life. The manuscript by Jasnin et al reports the cryoET structure of toroid assembly formation of FtsZ filaments driven by ZapD as the cross linker. The findings are **important** and have the potential to open a new dimension in the field, and the evidence to support these exciting claims is **solid**.

---

## [Referee Report · Reviewer #1 (Public review)]

Summary:

The major result in the manuscript is the observation of the higher order structures in a cryoET reconstruction that could be used for understanding the assembly of toroid structures. The cross-linking ability of ZapD dimers result in bending of FtsZ filaments to a constant curvature. Many such short filaments are stitched together to form a toroid like structure. The geometry of assembly of filaments - whether they form straight bundles or toroid like structures - depends on the relative concentrations of FtsZ and ZapD.

Strengths:

In addition to a clear picture of the FtsZ assembly into ring-like structures, the authors have carried out basic biochemistry and biophysical techniques to assay the GTPase activity, the kinetics of assembly, and the ZapD to FtsZ ratio.

Weaknesses:

Future scope of work includes the molecular basis of curvature generation and how molecular features of FtsZ and ZapD affect the membrane binding of the higher order assembly.

---

## [Referee Report · Reviewer #3 (Public review)]

Summary:

Previous studies have analyzed the binding of ZapD to FtsZ and provided images of negatively stained toroids and straight bundles, where FtsZ filaments are presumably crosslinked by ZapD dimers. Toroids without ZapD have also been previously formed by treating FtsZ with crowding agents. The present study is the first to apply cryoEM tomography, which can resolve the structure of the toroids in 3D. This shows a complex mixture of filaments and sheets irregularly stacked in the Z direction and spaced radially. The most important interpretation would be to distinguish FtsZ filaments from ZapD crosslinks, This is less convincing. The authors seem aware of the ambiguity: "However, we were unable to obtain detailed structural information about the ZapD connectors due to the heterogeneity and density of the toroidal structures, which showed significant variability in the conformations of the connections between the filaments in all directions." Therefore, the reader may assume that the crosslinks identified and colored red are only suggestions, and look for their own structural interpretations.

Strengths:

This is the first cryoEM tomography to image toroids and straight bundles of FtsZ filaments bound to ZapD. A strength is the resolution, which. at least for the straight bundles. is sufficient to resolve the ~4.5 nm spacing of ZapD dimers attached to and projecting subunits of an FtsZ filament. Another strength is the pelleting assay to determine the stoichiometry of ZapD:FtsZ (although this also leads to weaknesses of interpretation).

---

## [Author Response]

The following is the authors’ response to the previous reviews

**Reviewer #1 (Public review):**
Summary:The major result in the manuscript is the observation of the higher order structures in a cryoET reconstruction that could be used for understanding the assembly of toroid structures. The cross-linking ability of ZapD dimers result in bending of FtsZ filaments to a constant curvature. Many such short filaments are stitched together to form a toroid like structure. The geometry of assembly of filaments - whether they form straight bundles or toroid like structures - depends on the relative concentrations of FtsZ and ZapD.Strengths:In addition to a clear picture of the FtsZ assembly into ring-like structures, the authors have carried out basic biochemistry and biophysical techniques to assay the GTPase activity, the kinetics of assembly, and the ZapD to FtsZ ratio.Weaknesses:The discussion does not provide an overall perspective that correlates the cryoET structural organisation of filaments with the biophysical data. The current version has improved in terms of addressing this weakness and clearly states the lacuna in the model proposed based on the technical limitations.Future scope of work includes the molecular basis of curvature generation and how molecular features of FtsZ and ZapD affect the membrane binding of the higher order assembly.
**Reviewer #3 (Public review):**
Summary:Previous studies have analyzed the binding of ZapD to FtsZ and provided images of negatively stained toroids and straight bundles, where FtsZ filaments are presumably crosslinked by ZapD dimers. Toroids without ZapD have also been previously formed by treating FtsZ with crowding agents. The present study is the first to apply cryoEM tomography, which can resolve the structure of the toroids in 3D. This shows a complex mixture of filaments and sheets irregularly stacked in the Z direction and spaced radially. The most important interpretation would be to distinguish FtsZ filaments from ZapD crosslinks, This is less convincing. The authors seem aware of the ambiguity: "However, we were unable to obtain detailed structural information about the ZapD connectors due to the heterogeneity and density of the toroidal structures, which showed significant variability in the conformations of the connections between the filaments in all directions." Therefore, the reader may assume that the crosslinks identified and colored red are only suggestions, and look for their own structural interpretations. But readers should also note some inconsistencies in stoichiometry and crosslinking arrangements that are detailed under "weaknesses."Strengths.This is the first cryoEM tomography to image toroids and straight bundles of FtsZ filaments bound to ZapD. A strength is the resolution, which. at least for the straight bundles. is sufficient to resolve the ~4.5 nm spacing of ZapD dimers attached to and projecting subunits of an FtsZ filament. Another strength is the pelleting assay to determine the stoichiometry of ZapD:FtsZ (although this also leads to weaknesses of interpretation).WeaknessesThe stoichiometry presents some problems. Fig. S5 uses pelleting to convincingly establish the stoichiometry of ZapD:FtsZ. Although ZapD is a dimer, the concentration of ZapD is always expressed as that of its subunit monomers. Fig. S5 shows the stoichiometry of ZapD:FtsZ to be 1:1 or 2:1 at equimolar or high concentrations of ZapD. Thus at equimolar ZapD, each ZapD dimer should bridge two FtsZ's, likely forming crosslinks between filaments. At high ZapD, each FtsZ should have it's own ZapD dimer. However, this seems contradicted by later statements in Discussion and Results. (1) "At lower concentrations of ZapD, .. toroids are the most prominent structures, containing one ZapD dimer for every four to six FtsZ molecules." Shouldn't it be one ZapD dimer for every two FtsZ? (2) "at the high ZapD concentration...a ZapD dimer binds two FtsZ molecules connecting two filaments." Doesn't Fig. S5 show that each FtsZ subunit has its own ZapD dimer? And wouldn't this saturate the CTD sites with dimers and thus minimize crosslinking?

We thank the reviewer for these insightful comments. The affinity of ZapD for FtsZ is relatively low and a higher concentration of ZapD is required in solution to effectively saturate the binding sites of all FtsZ molecules forming macrostructures. It is important to clarify that the concentrations mentioned in the text refer to the amounts and ratios of protein added to the total volume of the sample, rather than the proteins actively interacting and forming bundles or macrostructures.

To differentiate, two aspects can be considered: the ratio of added protein (as mentioned in the text) and the fraction of proteins that contribute to the formation of the macrostructures. Under polymerization conditions, FtsZ-GTP recruits additional monomers to form polymers. Therefore, more FtsZ than ZapD would be involved in forming filaments and bundles. Our results support this hypothesis and show that a higher amount of ZapD is required in the sample to pellet with FtsZ bundles.

We propose that starting with the same initial concentration of FtsZ and ZapD in solution, only a small fraction of ZapD will bind to the structures, favoring the formation of toroidal structures despite the initial 1:1 ratio of proteins added to the sample. When considering a higher FtsZ:ZapD ratio (1:6), the increased amount of ZapD in solution would facilitate the saturation of all FtsZ binding sites, consistent with the observation of straight bundles. Analytical sedimentation velocity data further supported this finding, indicating a binding ratio of approximately 0.3-0.4, suggesting that one ZapD dimer binds for every 4-6 FtsZ monomers. The binding ratio indicates that two FtsZ monomers will bind to a single dimer of ZapD, but this only occurs when there is a significant excess of ZapD over FtsZ in the solution mixture.

These findings align qualitatively with the relative intensities of the electrophoretic bands observed for FtsZ and ZapD in the pelleting assay with different FtsZ-ZapD mixtures, as shown in Suppl. Fig. 5 as % of FtsZ in the fractions. Without prior staining calibration of the gels, there is no simple quantitative relationship between gel band intensities after Coomassie staining and the amount of protein in a band (Darawshe et al. 1993 Anal Biochem - DOI: 10.1006/abio.1993.1581). This last point precludes a quantitative comparison between pelleting/SDS-PAGE data and analytical sedimentation measurements. For this reason, we have decided to present pelleting results as % of FtsZ in supernatant and pellet to avoid overestimations.

A major weakness is the interpretation of the cryoEM tomograms, specifically distinguishing ZapD from FtsZ. The distinction of crosslinks seems based primarily on structure: long continuous filaments (which often appear as sheets) are FtsZ, and small masses between filaments are ZapD. The density of crosslinks seems to vary substantially over different parts of the figures. More important, the density of ZapD's identified and colored red seem much lower than the stoichiometry detailed above. Since the mass of the ZapD monomer is half that of FtsZ, the 1:1 stoichiometry in toroids means that 1/3 of the mass should be ZapD and 2/3 FtsZ. However, the connections identified as ZapD seem much fewer than the expected 1/3 of the mass. The authors conclude that connections run horizontally, diagonally and vertically, which implies no regularity. This seems likely, but as I would suggest that readers need to consider for themselves what they would identify as a crosslink.

The amount of ZapD in the toroids will be significantly less than one third. Although the theoretical addition of protein to the samples is at a 1:1 ratio, the actual amount of protein in the macrostructures containing ZapD is much lower, as shown by sedimentation velocity pelleting assays.

In contrast to the toroids formed at equimolar FtsZ and ZapD, thin bundles of straight filaments are assembled in excess ZapD. Here the stoichiometry is 2:1, which would mean that every FtsZ should have a bound ZapD DIMER. The segmentation of a single filament in Fig. 5e seems to agree with this, showing an FtsZ filament with spikes emanating like a picket fence, with a 4.5 nm periodicity. This is consistent with each spike being a ZapD dimer, and every FtsZ subunit along the filament having a bound ZapD dimer. But if each FtsZ has its own dimer, this would seem to eliminate crosslinking. The interpretative diagram in Fig. 6, far right, which shows almost all ZapD dimers bridging two FtsZs on opposite filaments, would be inconsistent with this 2:1 stoichiometry.

Assessing the precise stoichiometry of FtsZ and ZapD within the macrostructures is challenging. We interpret the spikes as ZapD dimers bridging two FtsZ filaments, implying a theoretical 1:1 stoichiometry in the straight bundle. However, ZapD may be enriched in certain areas, indicating that a single FtsZ monomer is binding to one side of the dimer. In contrast, the other side remains available for additional connections, resulting in a potential 2:1 stoichiometry. A combination of both scenarios is likely, although our resolution does not allow further characterization. Considering these complexities, we assume these connections represent a dimer of ZapD binding to two FtsZ monomers.

Figure 6 shows a simplified scheme illustrating how the bundles could be assembled based on the Cryo-ET data. We acknowledge the limitations of this diagram; its purpose is to depict the mesh formed by the stabilization of ZapD. We have not included interactions that do not lead to filament crosslinking, such as dimers binding to only one FtsZ filament. This focus enhances the interpretation of the scheme and the FtsZ-ZapD interaction. A sentence has been added to the caption to highlight the possibility of other interactions not considered in the scheme.

In the original review I suggested a control that might help identify the structures of ZapD in the toroids. Popp et al (Biopolymers 2009) generated FtsZ toroids that were identical in size and shape to those here, but lacking ZapD. These toroids of pure FtsZ were generated by adding 8% polyvinyl chloride, a crowding agent. The filamentous substructure of these toroids in negative stain seemed very similar to that of the ZapD toroids here. CryoET of these toroids lacking ZapD might have been helpful in confirming the identification of ZapD crosslinks in the present toroids. However, the authors declined to explore this control.

The mechanisms by which methylcellulose (MC) promotes the assembly of FtsZ macrostructures reported by Popp et al. involve more than simple excluded volume effects, as the low concentration of MC (less than 1 mg/ml) falls below the typical crowding regime. The latter suggests the existence of poorly characterized additional interactions between MC and FtsZ. These complexities preclude the use of FtsZ polymers formed in the presence of MC as a true control for the FtsZ toroidal structures reported here.

Finally, it should be noted that the CTD binding sites for ZapD should be on the outside of curved filaments, the side facing the membrane in the cell. All bound ZapD should project radially outward, and if it contacted the back side of the next filament, it should not bind (because the CTD is on the front side). The diagram second to right in Fig. 6 seems to incorporate this abortive contact.

The role of the flexible linker and its biological implications are still under debate in the field. The flexible linker allows ZapD-driven connections to be made in different directions. While these implications are not the primary focus of our manuscript, the flexible linker could allow connections between filaments in different orientations.

**Reviewer #1 (Recommendations for the authors):**
Most of the concerns which I had raised in the earlier version have been taken care of, as detailed in the response.A few minor points, mostly related to re-phrasing are listed below:Page 2: line 21: The use of the term 'C-terminal domain' for the C-terminal unstructured region of FtsZ is confusing. The term C-terminal domain or CTD for FtsZ is commonly used to describe part of the globular domain, while C-terminal tail or CCTP will be a more apt usage for all the instances in this manuscript.

We refer to the C-terminal domain as the carboxy-terminal region of the protein. This domain includes the C-terminal linker (CTL), which varies in length between species, followed by a conserved 11-residue sequence (CTC) and shorter, variable C-terminal sequences (CTV). We used the term "C-terminal domain" primarily to improve the readability of the manuscript, but we appreciate the reviewer's feedback. We have now adopted the term "CCTP" instead of "C-terminal domain" to improve the clarity of our manuscript.

On a related note, the schematic in Fig 1 shows the interaction with CCTP rather than the C-terminal domain of the globular FtsZ. Please provide an explanation.

We refer to the unstructured C-terminal domain of FtsZ as the C-terminal tail. To avoid confusion, we have introduced the term CCTP in this manuscript.

Supple Fig 2: "The FCS analysis demonstrated an increasing diffusion time of ZapD along with the FtsZ concentration as result of higher proportion of ZapD bound to FtsZ.The increased diffusion time need not be interpreted as increased ZapD bound, it could also mean that FtsZ could polymerise in the presence of increasing ZapD, was this possibility ruled out? Including a comment on this aspect will be useful.

In these experiments, we monitored fluorescently labeled ZapD. Due to their interaction, we found that its diffusion time increased at high FtsZ concentrations. The data presented in Supplementary Figure 2 shows ZapD in the presence of FtsZ-GDP (i.e. under non-polymerization conditions).

Was it possible to get a molecular weight estimate based on the diffusion time?

It is possible to estimate hydrodynamic volumes using the Stokes-Einstein equation if the diffusion coefficient of the diffusing particles is known, assuming that the particles are small and spherical. A molecular weight can then be estimated using a standard density of 1.35 g/cm3 (Fisher et all. Protein science 2009 DOI: 10.1110/ps.04688204). This estimate is heavily dependent on the shape of the diffusing particle, as we assume that our protein of interest here is far from a spherical shape due to the interaction through the flexible linker, the hydrodynamic volumes are overestimated. This overestimation then leads to a further overestimation of the molecular weight. In addition, for a more accurate estimation of the sizes and thus molecular weights for proteins, a modified model of the Stokes-Einstein equation is required (Tyn and Gusek Biotechnology and Bioengineering DOI: 10/1002/bit.260350402), where additional information about the shape of the diffusing particle is estimated by measuring the radius of gyration of the particle. These calculations are complex and beyond the scope of our manuscript.

Supple Fig 4:Does FtsZ GTPase activity (without ZapD) also vary with KCl concentrations? It will be useful to comment on this in Supplementary Figure 4.

Yes, it has been previously reported that moderate concentration of KCl is optimal for FtsZ GTPase activity. We added a comment to the caption.

Page 6, line 42: short filament segments arranged nearly 'parallel' to each other Since FtsZ filaments are polar, it is better to rephrase as 'parallel or antiparallel'.

Corrected.

Page 7, line 41: cross linking of short 'FtsZ' filaments and not ZapD?

It was a typo. Corrected

Page 8: delete 'from above' in the title?

Corrected

The use of the phrases such as 'cross linking from the top'; 'binds to FtsZ from above' is vague. (Figure 5b legend; discussion page 10, line 18; page 8, line 26; page 12, line 27). Similarly labelling on a schematic figure on the use of vertical, diagonal/lateral will be useful for the readers.

We thank the reviewer for the suggestions to improve the understanding of our data. We have simplified them by renaming these interactions as vertical.

Page 13, lines 6 -10Rather than an orientation of top or from the side, just the presence of multiple crosslinks along coaxial filaments suffices for a straight bundle. The average spacing will be more uniform in such a straight bundle compared to a toroid where there might be regions without ZapD. I do not find the data on an upward orientation convincing. ZapD binding need not be above to have the C-terminal ends of FtsZ pointing towards the membrane. On the other hand, having ZapD bind above is likely to occlude membrane binding of FtsZ?

The flexibility of the FtsZ linker suggests that ZapD can bind filaments oriented in different directions. In a cellular environment, FtsZ molecules interact with other division proteins that compete with ZapD for binding sites. This competition could prevent the membrane from occluding and instead create binding sites between the filaments, stabilizing them.

Page 11, lines 32 - 34: Please rephrase the sentence, with focus on the main point to be conveyed. Do the authors want to say that the 'Same molecule contributes to variability in spacing based on the number of connections formed.'

Thank you for your comment. We have rephrased the sentence for clarity.

Page 11: paragraphs 1,2, and 3 appears to convey similar, related ideas and are redundant. Could these be shortened further into one paragraph highlighting how the ratio leads to differences in higher order FtsZ organisation?

These paragraphs discuss different ideas, and it is better to keep them separate.

In the response to reviewers, page 19, point 5 (iii), it is given that 5000 FtsZ molecules correspond to 2/3rd of the total, while in the manuscript text, it is given as one-third. Please correct the response text/manuscript text accordingly. The numbers in the cited reference appears to suggest 1/3rd.

Yes, it was 1/3rd. Thanks for pointing that out.

Fig 1b. Y-axis: Absorbance spelling has a typo.Page 14, line 11: Healthcare ('h' missing)Page 14, line 15: HCl, KCl (L should be in small letter)Page15, line 18: 43 - 48K rpm (not Krpm)Supple Fig 1 legend: line 5: 's' missing for species

Corrected.